# Context Tokens are Anchors: Understanding the Repeat Curse in dMLLMs from an Information Flow Perspective

**Qiyan Zhao**[1,2,3*] , **Xiaofeng Zhang**[1*‡] , **Shuochen Chang**[1] , **Qianyu Chen**[4] , **Xuhang Chen**[1]
**Xiaosong Yuan**[1] , **Luoqi Liu**[5] , **Jiajun Zhang**[3] , **Xu-Yao Zhang**[3] , **Da-Han Wang**[2‡]

[1]SJTU    [2]XMUT    [3]CASIA    [4]NTU    [5]Meitu (China) Limited
[*]Equal contribution    [‡]Corresponding author

## Abstract

Recent diffusion-based Multimodal Large Language Models (dMLLMs) suffer from high inference latency and therefore rely on caching techniques to accelerate decoding. However, the application of cache mechanisms often introduces undesirable repetitive text generation, a phenomenon we term the **Repeat Curse**. To better investigate underlying mechanism behind this issue, we analyze repetition generation through the lens of information flow. Our work reveals three key findings: (1) context tokens aggregate semantic information as anchors and guide the final predictions; (2) as information propagates across layers, the entropy of context tokens converges in deeper layers, reflecting the model's growing prediction certainty; (3) Repetition is typically linked to disruptions in the information flow of context tokens and to the inability of their entropy to converge in deeper layers. Based on these insights, we present **CoTA**, a plug-and-play method for mitigating repetition. CoTA enhances the attention of context tokens to preserve intrinsic information flow patterns, while introducing a penalty term to the confidence score during decoding to avoid outputs driven by uncertain context tokens. With extensive experiments, CoTA demonstrates significant effectiveness in alleviating repetition and achieves consistent performance improvements on general tasks. Code is available at `https://github.com/ErikZ719/CoTA`.

## 1 Introduction

Recent advances in diffusion-based large language models (dLLMs) Nie et al. (2025b); Zhu et al. (2025a); Ye et al. (2025a) have demonstrated impressive reasoning and parallel decoding capabilities. Unlike autoregressive large language models Touvron et al. (2023a); Yang et al. (2024a), which generate text by sequentially predicting the next token, dLLMs produce tokens in parallel by framing response generation as an iterative denoising process over a fully masked discrete token sequence. Building on this foundation, diffusion-based Multimodal Large Language Models (dMLLMs) You et al. (2025); Yang et al. (2025b); Li et al. (2025b) have emerged as powerful multimodal systems that integrate vision-instruction tuning with dLLMs, and have evolved into promising alternatives to autoregressive architectures Liu et al. (2023a); Li et al. (2025a).

Current dMLLMs face notable inference latency, stemming from their multi-step denoising procedure and the use of bidirectional attention. Existing efforts Liu et al. (2025b); Wei et al. (2025a); Wu et al. (2025) mainly exploit customized caching strategies for both prefix and suffix tokens to accelerate dMLLM inference. However, our experiments reveal that these methods often introduce a severe side effect: the generated text exhibits substantial repetition, as illustrated in Figure 1, a phenomenon we refer to as the **Repeat Curse**. This repetition significantly reduces the performance and readability of the model outputs. We conduct a statistical analysis of the repeat curse across different dMLLMs and caching techniques, as shown in Figure 2. The empirical results reveal the widespread presence of the repeat curse when dMLLMs are equipped with current caching methods.

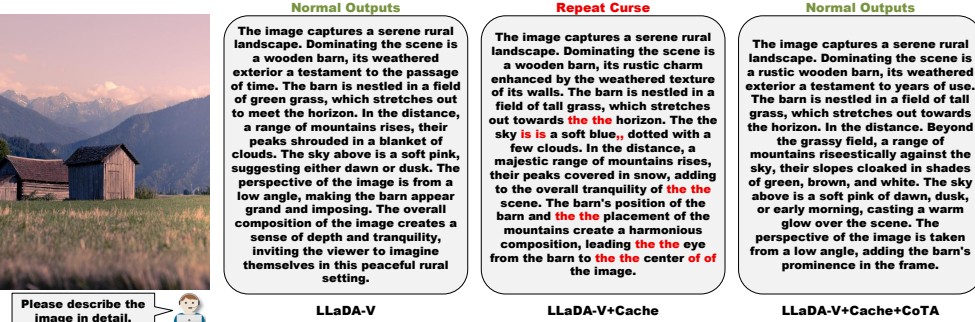

Figure 1: **Motivation.** When cache is applied to accelerate dMLLMs, the generated responses often exhibit excessive token repetition—a phenomenon we term the Repeat Curse.

However, the inherent black-box nature of dMLLMs poses a challenge to uncovering the underlying mechanisms behind the repeat curse. Recent advances in information flow Yu et al. (2024); Wang et al. (2023); Chen et al. (2024a) have introduced an interpretable analysis paradigm for understanding the relationship between model outputs and internal mechanisms, which has motivated the development of numerous methods for mitigating abnormal generation behaviors, such as hallucination Tang et al. (2025b) and degeneration Yona et al. (2025). Inspired by these successes, this work conducts an in-depth analysis from the perspective of information flow to reveal the connection between the internal mechanisms of dMLLMs and the repeat curse.

By visualizing the attention interaction patterns among tokens (as shown in Figure 3), we observe that in dMLLMs, the context tokens adjoining the query act as anchors that progressively aggregate semantic information across layers, **thereby causing attention distribution to gradually concentrate on these context tokens**. We further introduce information entropy to quantitatively analyze the impact of context tokens on the decoding dynamics, as illustrated in Figure 4 (a). We find that **the entropy of context tokens typically converges in deeper layers**, indicating that as information is progressively aggregated across the layers, the model's predictive certainty consistently increases. However, when repetition arises under caching mechanisms (e.g., dLLM-Cache or SlowFast), the model tends to exhibit a highly random attention distribution, as illustrated in Figure 3. Moreover, the generated context tokens with repetition during decoding exhibit persistently abnormally high entropy in deeper layers, as shown in Figure 4. We compute the per-layer entropy over multiple samples and report the averaged results in Figure 5, where the resulting statistics further validate the above findings. Together, these findings suggest that caching disrupts the inherent mechanisms of dMLLMs, thereby triggering the repeat curse. We attribute this phenomenon to two key factors: 1. The introduction of caching disrupts the attention distribution and the inherent information flow patterns of context tokens; 2. Some context tokens whose entropy fails to converge in deeper layers induce the model to generate uncertain tokens, which are often accompanied by repetition.

Building on these insights, we propose **CoTA**[1], a plug-and-play approach that addresses the abnormal patterns underlying the repeat curse and mitigates repetition. CoTA is built on two key components: (1) Context-token Attention Enhancement (CTAE): a distance-aware attention intervention that strengthens attention to context tokens, thereby preserving the intended information flow during token interactions; (2) Context-token Entropy-guided Voting (CTEV): a mechanism that leverages the aggregated deep-layer entropy of context tokens as a penalty term in the confidence score, discouraging the model from generating uncertain and repetitive outputs. CoTA can be seamlessly integrated with baseline dMLLMs and existing caching strategies in a training-free manner, while incurring only modest computational overhead. The experimental results demonstrate the effectiveness of CoTA in mitigating the repetition curse, as well as its robustness across different dMLLMs and caching methods. Moreover, across several multimodal benchmarks, CoTA consistently achieves performance gains over the baseline models, further validating its generalization. Our contributions are three-fold:

- First, we identify the repeat curse that emerges when caching is applied in dMLLMs and uncover its underlying causes through information flow analysis.

---

[1]The name **CoTA** is derived from our key finding: **Co**ntext **T**okens are **A**nchors.

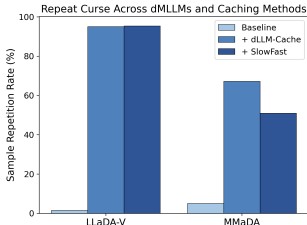 

Figure 2: The statistical analysis of the repeat curse on LLaDA-V and MMaDA.

Figure 3: The information-flow visualizations from left to right correspond to LLaDA-V, LLaDA-V+dLLM-Cache Liu et al. (2025b), and LLaDA-V+SlowFast Wei et al. (2025a).

- Second, we introduce CoTA, a plug-and-play method specifically designed to alleviate repetition.
- Third, we validate the effectiveness of CoTA through extensive experiments, demonstrating consistent performance improvements on general multimodal tasks.

## 2 RELATED WORK

**dMLLMs and Caching Methods.** The latest dLLMs Nie et al. (2025b); Zhu et al. (2025a); Ye et al. (2025a); Gong et al. (2025) have been successfully scaled to 8B parameters, achieving performance comparable to state-of-the-art autoregressive large language models (LLMs) Grattafiori et al. (2024); DeepSeek-AI et al. (2024); Yang et al. (2025a). By introducing visual encoders Tschannen et al. (2025); Bai et al. (2025) into dLLMs and incorporating visual instruction tuning Liu et al. (2023a), recent studies You et al. (2025); Yang et al. (2025b); Li et al. (2025b); Yu et al. (2025) have successfully developed dMLLMs. Despite their strong performance across diverse tasks, high inference latency remains a major bottleneck, underscoring the need for effective acceleration methods. In autoregressive models, KV caching Pope et al. (2023) has been widely adopted to accelerate inference by reusing intermediate representations. However, due to the bidirectional, non-causal attention in diffusion models, conventional caching techniques are difficult to apply directly. Recent studies, such as Fast-dLLM Wu et al. (2025), Prefix-DLM Li et al. (2025b), dLLM-Cache Liu et al. (2025b), and SlowFast Wei et al. (2025a), have successfully adapted the caching paradigm to dLLMs and dMLLMs. This work reveals a critical limitation of these caching approaches: the repeat curse.

**Information Flow.** Recent research on information flow has provided an interpretable approach for probing the internal mechanisms of black-box models. Common approaches mainly include saliency scores Zhang et al. (2025); Wang et al. (2023); Huo et al. (2025); Zhuang et al. (2025); Deng et al. (2025), attention maps Tong et al. (2025b); Huang et al. (2023); Zhao et al. (2025c); Kang et al. (2025); Qiu et al. (2025); Song et al. (2025a); Chen et al. (2025); Ye et al. (2025b), Grad-CAM Zhang et al. (2024b); Li et al. (2025c); Wang et al. (2025b); Xing et al. (2025), information entropy Yao et al. (2025); Nguyen et al. (2025), and massive values Jin et al. (2025); An et al. (2025); Kaul et al. (2025); Gan et al. (2025a); Gu et al. (2025); Gan et al. (2025b), among others Zou et al. (2025). Through information-flow analysis, prior works have observed that, in autoregressive MLLMs, certain tokens act as "anchors" to receive disproportionately high attention, and have further analyzed their impact on model reasoning Zhang et al. (2024c); Chen et al. (2024a) and hallucination Tang et al. (2025a); Kang et al. (2025). In contrast, this work presents the in-depth information-flow analysis of dMLLMs to uncover the underlying mechanism behind the repeat curse.

**Token Repetition in Generation.** Early research on token repetition primarily focused on n-gram penalties Zhu et al. (2023), decoding strategies Su et al. (2022); Fu et al. (2021), and training-time optimization Lagutin et al. (2021); Xu et al. (2022). More recent studies have extended the investigation to autoregressive LLMs Li et al. (2023b); Yang et al. (2024b); Ginart et al. (2025). However, these approaches largely focus on mitigation rather than explaining the underlying causes of repetition. In contrast, DUC Yao et al. (2025) and model editing Wang et al. (2024) analyze the origins of repetition in LLMs from the perspectives of feature and neurons, respectively. To the best of our knowledge, we are the first to reveal **the underlying causes of token repetition in dMLLMs from an information-flow perspective**.

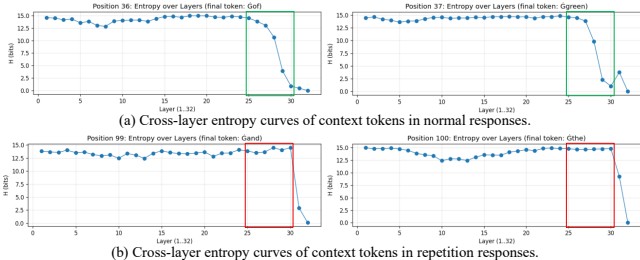
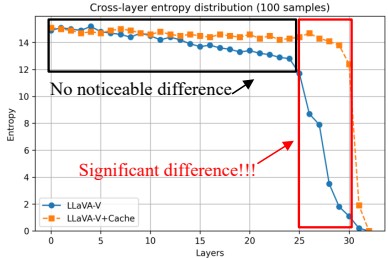

Figure 4: (a) and (b) respectively show the cross-layer entropy curves of local context tokens under non-repetitive and repetitive cases. We observe that neighboring tokens without repetition exhibit a sharp decay in deep-layer entropy, whereas neighboring tokens with repetition tend to maintain persistently high entropy in deeper layers.

Figure 5: Statistical results of cross-layer entropy distributions across multiple samples (randomly selected from the COCO Val Lin et al. (2014)). We compute the mean entropy at each layer.

## 3 MOTIVATION AND ANALYSIS

In this section, we begin with a brief overview of baseline dMLLMs and the cache mechanism[2]. We then investigate the repeat curse through both quantitative and qualitative analyses. Finally, we compare the information flow in dMLLMs with and without cache, shedding light on the underlying cause of the repeat curse.

### 3.1 PRELIMINARY

**Architecture of dMLLMs** Typically, a dMLLM $\mathcal{F}$ consists of three main components: a pretrained vision encoder $\mathcal{F}_v$, a dLLM $\mathcal{F}_t$, and a projector $f$ that maps visual features into the text embedding space. Given an image input $I_v$ and an instruction prompt input $I_p$ (e.g., *"Please describe the image in detail"*), the model converts them into tokens and concatenates them into a multimodal sequence: $\{\mathcal{S}_v, \mathcal{S}_t\}$, where $\mathcal{S}_v = f(\mathcal{F}_v(I_v)) = \{w_i\}_{i=1}^V$ and $\mathcal{S}_t = \mathcal{F}_t(I_p) = \{w_i\}_{i=1}^T$ represent visual and instruction tokens of lengths $V$ and $T$, respectively. Subsequently, a fully masked sequence $\mathcal{S}_m = \{w_i\}_{i=1}^M$, where $w_i = [\text{MASK}]$ and $[\text{MASK}]$ denotes the special mask token, is initialized as the starting state of the response sequence with maximum generation length $M$. Finally, $\mathcal{S}_m$ is concatenated with $\{\mathcal{S}_v, \mathcal{S}_t\}$ to form the complete sequence $\mathcal{S} = \{\mathcal{S}_v, \mathcal{S}_t, \mathcal{S}_m\}$, which is fed into the dLLMs for iterative denoising. In this sequence, all tokens share the same dimensionality.

**Inference of dMLLMs.** Let $\mathcal{S}^t = \{\mathcal{S}_v^t, \mathcal{S}_t^t, \mathcal{S}_m^t\}$ denote the input sequence state at decoding step $t$, where $t \in \{N, N-1, \ldots, 1\}$. Starting from $\mathcal{S}^N = \mathcal{S}$, where the response sequence $\mathcal{S}_m$ is fully masked, the model performs iterative denoising of $\mathcal{S}_m^t$ over $N$ discrete steps to predict the final clean response sequence $\mathcal{S}_m^0$, progressing from $k = N$ to $k = 1$. At each step $t$, the model estimates a probability distribution $p_\theta(\mathcal{S}^0 \mid \mathcal{S}^t)$ for each masked token based on the noisy state $\mathcal{S}^t$. Subsequently, the prediction of original clean sequence at step $t$ is obtained via greedy decoding as:

$$\hat{\mathcal{S}}_{(i)}^t = \arg\max_{v \in V} p_\theta(\mathcal{S}_{(i)}^0 = v \mid \mathcal{S}^t) \quad \forall i \in \{1, \ldots, M\}, \tag{1}$$

where $V$ is the token vocabulary. Although the model can predicts all masked tokens at each step, to enable finer-grained decoding, dMLLMs adopt a remasking strategy $\mathcal{R}$ to the unmasked tokens. Therefore, to obtain the next state $\mathcal{S}^{t-1}$ from $\hat{\mathcal{S}}^t$, an unmasking process is required (e.g., random or by confidence). Taking the confidence-based remasking strategy as an example[3], the corresponding confidence scores $c_{(i)}$ based on $\hat{\mathcal{S}}_{(i)}^t$ can be obtained as:

$$c_{(i)} = p_\theta(\mathcal{S}_{(i)}^0 = \hat{\mathcal{S}}_{(i)}^t \mid \mathcal{S}^t), \quad i \in \mathcal{I}_{mask}^t, \tag{2}$$

where $\mathcal{I}_{mask}^t$ represents the index set of masked token positions at step $t$. Finally, the model selects the $k$ positions in $\mathcal{I}_{mask}^t$ with the highest confidence scores $c_{(i)}$ and obtains the set of indices $\mathcal{I}_{update}^t$

---

[2]We adopt LLaDA-V You et al. (2025) and dLLM-Cache Liu et al. (2025b) as the baseline dMLLMs and cache method.

[3]The discussion of the random remasking strategy is provided in the Appendix C.

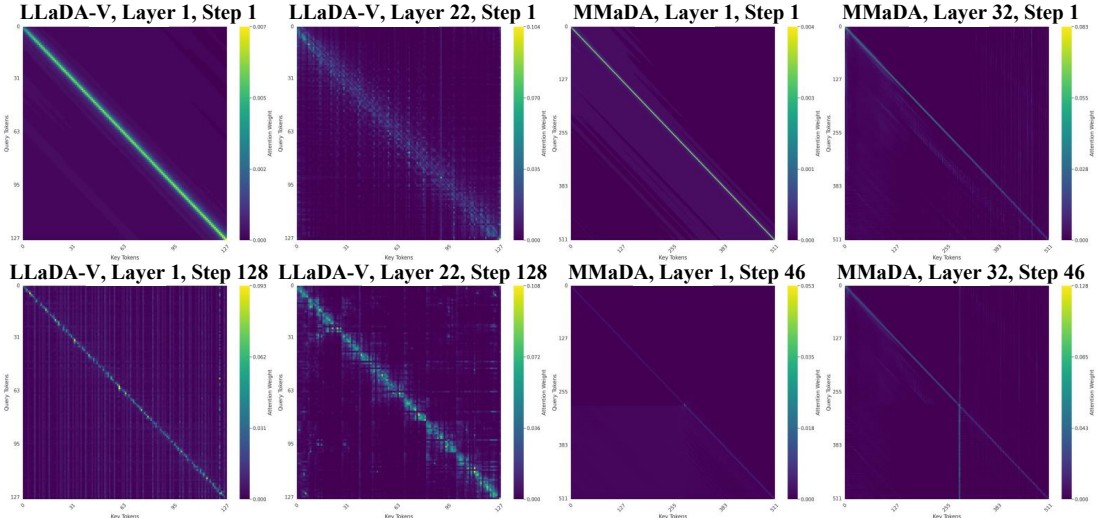

Figure 6: **Context Tokens are Anchors**. We visualize the attention maps of two dMLLMs, LLaDA-V and MMaDA, to analyze their information-flow patterns. The x-axis and y-axis correspond to key and query tokens, respectively, where brighter colors indicate higher attention weights during query–key interactions. For LLaDA-V, the maximum generation length is 128, while for MMaDA it is 512. We present attention maps across different depths and decoding steps, with additional results provided in Appendix B. We observe that neighboring context tokens act as anchors that aggregate information across layers and absorb a disproportionate amount of attention.

to update. The $k$ selected tokens are unmasked at this step, producing the updated sequence:

$$\mathcal{S}_{(i)}^{t-1} = \begin{cases} \hat{\mathcal{S}}_{(i)}^{t}, & \text{if } i \in \mathcal{I}_{update}^{t}, \\ \mathcal{S}_{(i)}^{t}, & \text{otherwise}. \end{cases} \quad (3)$$

The indices of the masked tokens at step $t-1$ are updated by $\mathcal{I}_{mask}^{t-1} = \mathcal{I}_{mask}^{t} - \mathcal{I}_{update}^{t}$.

**Caching in dMLLMs.** During the iterative unmasking process of dMLLMs, the states of all tokens must be recomputed at each step, leading to significant inference latency. Unlike the causal attention in autoregressive models, which allows direct reuse of past KV states Xiao & et al. (2024), dMLLMs adopt bidirectional attention over all tokens, which fundamentally limits the reuse of intermediate states. Recent works Liu et al. (2025b); Huang et al. (2025); Wei et al. (2025a) reveal a similar observation: the features of prompt tokens $\{\mathcal{S}_v, \mathcal{S}_t\}$ and parts of the output tokens $\mathcal{S}_m$ across adjacent decoding steps exhibit high similarity, suggesting opportunities for token-state caching and reuse. Based on this insight, caching in dMLLMs can be organized according to token types as follows:

$$\begin{cases} \text{Recompute}(\{\mathcal{S}_v^t, \mathcal{S}_t^t\}), \text{if } t \bmod \mathcal{E}_p = 0, \\ \text{Recompute}(\mathcal{S}_m^t), \text{if } t \bmod \mathcal{E}_o = 0, \\ \text{Recompute}(\mathcal{S}_{(i)}^t), \text{if } \mathcal{S}_{(i)}^t \in \arg\min_\alpha \left( \text{Sim}\left(\mathcal{S}_m^t, \mathcal{S}_m^{t-1}\right) \right), \end{cases} \quad (4)$$

where $\text{Recompute}()$ denotes the recomputation. For prompt tokens, their states are recomputed every $\mathcal{E}_p$ steps, and the cached states are reused within each iteration interval. For output tokens, the set of tokens to be recomputed is determined according to two principles: First, similar to prompt caching, token states are recomputed periodically according to a fixed interval $\mathcal{E}_o$. Second, the cosine similarity between token features at adjacent steps is computed via $\text{Sim}()$, and the bottom $\alpha$ proportion of tokens, whose feature vectors vary most significantly, are selected for recomputation.[4]

---

[4]Unless otherwise specified, all hyperparameters are fixed to $\alpha = 0.25$, $\mathcal{E}_p = 25$, and $\mathcal{E}_o = 7$ to ensure fair comparison.

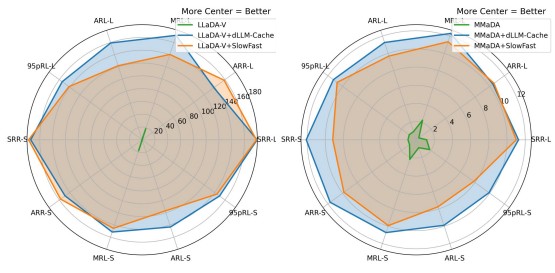

Figure 7: Radar charts of the repeat curse, where L and S denote long and short responses, respectively.

Table 1: Ablation study on cache mechanism design components.

| Interval | 1 | 5 | 15 | 25 |
|---|---|---|---|---|
| **SRR↓** | 0 | 0 | 0 | 0 |

(a) Prompt interval.

| Interval | 1 | 3 | 7 |
|---|---|---|---|
| **SRR↓** | 0 | 79.9 | 89.7 |

(b) Output interval.

| Threshold | 0 | 0.25 | 0.75 |
|---|---|---|---|
| **SRR↓** | 89.7 | 75.0 | 29.7 |

(c) Similarity thresholds.

| Policy | Prefix | dLLM-Cache |
|---|---|---|
| **SRR↓** | 0 | 75.0 |

(d) Reuse policies.

## 3.2 REPEAT CURSE

**Quantitative Metrics.** As illustrated in Figure 1, we define the redundant tokens repetition in model responses as the Repeat Curse. To quantitatively evaluate this phenomenon, we introduce the Adjacent Repetition Rate (ARR), which measures the proportion of repeated tokens within a response sequence $\{y_i\}_{i=1}^{M}$ of lengths $M$:

$$\text{ARR} = \frac{1}{M-1} \sum_{i=1}^{M-1} \mathbf{1}\left(y_i = y_{i+1}\right),\tag{5}$$

where $\mathbf{1}()$ is an indicator function that takes the value 1 when the condition is satisfied and 0 otherwise. In addition, we adopt the sample repetition rate (SRR) to quantify the proportion of samples exhibiting repetition in the test set:

$$\text{SRR} = N_R/N,\tag{6}$$

where $N$ denotes the total number of samples in the test set, and $N_R$ denotes the number of samples whose responses contain repeated tokens (i.e., $\text{ARR} > 0$). To further assess the severity of repetition, we additionally introduce three metrics: maximum repetition length (MRL), average repetition length (ARL), and the 95th-percentile repetition length (95pRL). The detailed computation procedures are presented in Appendix D.

**Quantitative Analysis of Repeat Curse.** Figure 7 presents the quantitative results of the Repeat Curse on two dMLLMs under both long-response and short-response settings. The experiments are conducted on a VQA task using 500 randomly sampled images from MSCOCO Lin et al. (2014). We observe that, after applying two advanced caching techniques, both LLaDA-V and MMaDA exhibit catastrophic token repetition. Furthermore, we ablate three key components of the cache mechanism (prompt recomputation interval, output recomputation interval, and similarity threshold) to analyze the underlying causes of repetition. As shown in Table 1(a), starting with the prompt token recomputation interval while fixing the output token recomputation interval to 1 and setting the similarity threshold to 0, we observe that periodically caching prompt tokens has negligible impact on repetition. Next, by fixing the prompt interval to 25 and the similarity threshold to 0, we find that varying the output interval leads to severe token repetition (Table 1(b)). In contrast, increasing the similarity threshold gradually mitigates the repetition (Table 1(c)). Finally, we compare dLLM-Cache with prefix KV cache Li et al. (2025b), which reuses only prompt tokens, to analyze the impact of different reuse policies. Across all experiments, we find that token repetition is primarily driven by the caching of output tokens, with a more comprehensive analysis provided in Appendix E.2.

## 3.3 INFORMATION FLOW ANALYSIS

As mentioned earlier, applying caching techniques leads to severe repeat curse in dMLLMs. To uncover the underlying mechanisms driving repeated token generation, we analyze and interpret this phenomenon from an information-flow perspective. For each output token in the sequence, we visualize its attention matrix to analyze the information-flow patterns among tokens during the interaction process.

As shown in Figure 6, neighboring context tokens consistently receive high attention throughout the decoding process. The positions at which these tokens attract high attention remain stable across lay-

ers and decoding steps, predominantly corresponding to adjacent query–key pairs. This information-flow pattern is consistently observed across different dMLLMs, including LLaDA-V and MMaDA. Interestingly, this behavior resembles the "attention sink" phenomenon reported in autoregressive models Chen et al. (2024a); Wang et al. (2023); Zhang et al. (2024b), where certain special tokens act as anchors that aggregate information and absorb disproportionately high attention. Under the bidirectional attention mechanism of dMLLMs, these context tokens similarly function as anchors that guide and shape the model's information interactions.

> **Finding 1: In dMLLMs, context tokens serve as anchors that aggregate information and absorb attention across layers and guide the final prediction.**

We further plot the cross-layer entropy curves of neighboring context tokens under normal decoding, as shown in Figure 4(a). We observe high entropy in the shallow layers, followed by a gradual convergence as depth increases—with a particularly sharp drop around layers 26–30. This indicates that as information is aggregated across layers, the model's prediction for the final output becomes increasingly stable and confident. This observation is also consistent with findings reported in prior studies Ali et al. (2025).

> **Finding 2: For context tokens under normal decoding, the information entropy gradually converges in the deeper layers.**

However, after applying caching, the model exhibits patterns that differ markedly from the above observations, as shown in Figure 3 and Figure 4(b). Specifically, caching leads to a randomized attention distribution, disrupting the original information flow from context tokens. Moreover, context tokens involved in repetition exhibit persistently high entropy in deeper layers, failing to converge as normal context tokens do during normal tokens decoding.

> **Finding 3: Repetition is typically linked to disruptions in the information flow of context tokens and to the inability of their entropy to converge in deeper layers.**

## 4 METHOD

Building on the findings in Section 3, we propose CoTA, a training-free method to mitigate repeat curse, which has two key components: (1) Context tokens attention enhancement, which preserves the intrinsic information flow pattern of context tokens, and (2) Context tokens entropy-guided voting, which prevents the decoding of uncertain tokens arising from non-convergent deep-layer entropy in context tokens.

### 4.1 CONTEXT TOKENS ATTENTION ENHANCEMENT (CTAE)

As analyzed in Section 3, context tokens in dMLLMs typically serve as anchors to aggregate information and absorb attention, but the use of cache disrupts this information flow pattern. We hypothesize that tokens in close proximity exhibit stronger semantic correlations; therefore, this context-token–driven attention pattern arises naturally to promote local semantic coherence and prevent abnormal repetition caused by attention being assigned to random tokens. Therefore, we propose Context Tokens Attention Enhancement (CTAE), as shown in Figure 8, whose key idea is to introduce a distance-based decay term that intervenes in the attention matrix element-wise. Given a query token $q_i$ at position $i$ and a key token $k_j$ at position $j$, the decay term $\mathcal{G}_{i,j}$ for each query–key pair is com-

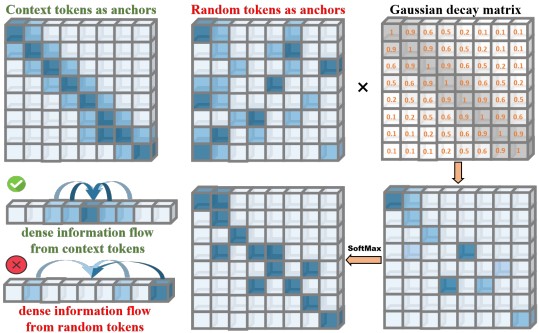

Figure 8: **Illustration of context tokens attention enhancement**. Example values are annotated on the decay matrix for clarity, while actual values are computed using Equation 7 and 8. The workflow of the CTAE algorithm is provided in Appendix E.4.

puted as follows:

$$\mathcal{G}_{i,j} = \gamma_{\min} + (1 - \gamma_{\min})g_{i,j}, \quad \gamma_{\min} \in (0, 1], \tag{7}$$

$$g_{i,j} = \exp\left(-\left(\frac{|i-j|}{\tau}\right)^2\right), \tag{8}$$

where $g_{i,j}$ denotes the Gaussian decay term computed by the exponential function $\exp()$ and the relative position distance $|i-j|$ between query–key pairs. $\tau$ denotes the temperature factor, which is fixed to 5 in the experiments. We introduce a lower-bound constant $\gamma_{\min}$ for stabilization, yielding the final decay term $\mathcal{G}_{i,j}$. The $\mathcal{G}_{i,j}$ values for all query–key pairs together form the Gaussian decay matrix shown in Figure 8. Assume the attention between a query token $q_i$ and a key token $k_j$ is $Attn_{i,j}$. We enhance the attention to context tokens by applying $Attn_{i,j} * \mathcal{G}_{i,j}$, thereby preserving the native information flow pattern of context tokens in dMLLMs.

## 4.2 CONTEXT TOKENS ENTROPY-GUIDED VOTING (CTEV)

As observed in Section 3, neighboring context tokens associated with repetition typically exhibit persistently high entropy in deeper layers, reflecting the model's uncertainty toward these tokens during decoding. Yet baseline dMLLMs rely solely on confidence scores to vote for candidate decoding tokens, ignoring this uncertainty. To address this, we propose Context Tokens Entropy-Guided Voting (CTEV), which incorporates the aggregated deep-layer entropy of context tokens as a penalty term on the confidence score to prevent decoding under uncertain context tokens. The complete algorithm is formalized in Algorithm **??**. First, we compute the entropy $E$ of each candidate token based on its softmax probability distribution $p_v$ as follows:

---

**Algorithm 1** Context Tokens Entropy-Guided Voting

**Require:** Deep-layer logits $z_{i,:}^{(l)}$, base confidences $c_{(i)}$, coefficient $\alpha$
1: **for** $i = 1$ to $M$ **do**
2:     $E_{\text{sum}}(i) \leftarrow 0$
3:     **for** $l \in \{26, \dots, 30\}$ **do**
4:         $p_v^{(l)} \leftarrow \exp(z_{i,v}^{(l)})/\sum_{u=1}^V \exp(z_{i,u}^{(l)})$
5:         $E_{\text{sum}}(i) \leftarrow E_{\text{sum}}(i) - \frac{1}{\log V}\sum_{v=1}^V p_v^{(l)} \log p_v^{(l)}$
6:     **end for**
7: **end for**
8: Build $\mathcal{C}(t) = \{t\} \cup \{$context tokens of candidate $t\}$
9: **for** $i \in \mathcal{C}(t)$ **do**
10:     $E_{\text{sum}}^{\text{ctx}}(i) \leftarrow \sum_{j \in \mathcal{C}(i)} E_{\text{sum}}(j)$
11:     $\text{Score}(i) \leftarrow c_{(i)} + \alpha \cdot E_{\text{sum}}^{\text{ctx}}(i)$
12: **end for**
13: **return** context tokens entropy-guided voting score

---

$$E = \frac{-\sum_{v=1}^V p_v \log p_v}{\log V}, p_v = \frac{\exp(z_v)}{\sum_{u=1}^V \exp(z_u)}, \tag{9}$$

where $z_v$ denotes the logits of the $v$-th word in the vocabulary, and $V$ is the vocabulary size. On this basis, the entropy of candidate decoding tokens in deeper layers[5] is accumulated layer by layer as follows:

$$E_{\text{sum}} = \sum_{l=26}^{30} E^{(l)} = \sum_{l=26}^{30} \frac{-\sum_{v=1}^V p_v^{(l)} \log p_v^{(l)}}{\log V}. \tag{10}$$

Let $E_{\text{sum}}(i)$ denote the $E_{\text{sum}}$ of token $x_i$ at position $i$ in a sequence of length $M$. The accumulated deep-layer entropy of the context tokens, $E_{\text{sum}}^{ctx}(i)$, is then computed as follows:

$$E_{\text{sum}}^{ctx}(i) = \sum_{j \in \mathcal{C}(i)} E_{\text{sum}}(i), \tag{11}$$

$$\mathcal{C}(i) = \{x_i\} \cup \left\{ x_j \mid j \in \underset{\{i,j\} \in [M]}{\arg\min_2} |j - i| \right\}. \tag{12}$$

Finally, the weighted $E_{\text{sum}}^{ctx}(i)$ with coefficient $\alpha$ is added into the confidence score $c_{(i)}$ from Equation 2, resulting in the new candidate tokens voting score $\text{Score}(i)$ as follows:

$$\text{Score}(i) = c_{(i)} + \alpha E_{\text{sum}}^{ctx}(i). \tag{13}$$

---

[5]We define layers 26–30 as the deep layers, and candidate tokens together with two nearest tokens as the context tokens.

## 5 EXPERIMENTS

### 5.1 EXPERIMENTAL SETUP DETAILS

Our baseline dMLLMs are LLaDA-V You et al. (2025) and MMaDA Yang et al. (2025b), and the baseline caching method is dLLM-Cache Liu et al. (2025b). The comparison methods include Fast-dLLM Wu et al. (2025), prefix KV Cache Li et al. (2025b), and SlowFast Wei et al. (2025a). The repeat curse is evaluated on the VQA task of the COCO dataset Lin et al. (2014), while general capabilities are assessed on DocVQA Mathew et al. (2020), ChartQA Masry et al. (2022), MME Fu et al. (2025), Seed Li et al. (2023a), MMBench Liu et al. (2024), MMMU Yue et al. (2024), LLaVA$^w$ Liu et al. (2023a) and MathVista Lu et al. (2024).

Table 2: **Repeat Curse Evaluation Results.** 512 and 64 denote the maximum generation length.

| Method | 512 | | | 64 | | |
|---|---|---|---|---|---|---|
| | ARR↓ | SRR↓ | len | ARR↓ | SRR↓ | len |
| LLaDA-V | 0.2 | 6.9 | 438.5 | 0.1 | 3.3 | 61.4 |
| + dLLM-Cache | 14.3 | 82.3 | 480.0 | 7.1 | 65.6 | 62.1 |
| + dLLM-Cache + CTAE | 3.2 | 10.6 | 469.3 | 2.5 | 5.6 | 60.5 |
| + dLLM-Cache + CTEV | 2.9 | 8.0 | 404.2 | 1.8 | 4.6 | 61.6 |
| + dLLM-Cache + CTEV + CTAE | 1.2 | 6.3 | 414.2 | 1.0 | 3.0 | 59.3 |

Table 3: Repeat curse across different settings.

| Method | ARR↓ | SRR↓ | | Method | ARR↓ | SRR↓ |
|---|---|---|---|---|---|---|
| MMaDA | 0.7 | 6.0 | | LLaDA-V | 0.1 | 3.3 |
| + dLLM-Cache | 4.4 | 55.0 | | + SlowFast | 7.4 | 69.2 |
| + dLLM-Cache + CTAE | 2.4 | 29.0 | | + SlowFast + CTAE | 2.7 | 7.4 |
| + dLLM-Cache + CTEV | 0.8 | 35.0 | | + SlowFast + CTEV | 1.9 | 6.2 |
| + dLLM-Cache + CoTA | 0.6 | 30.0 | | + SlowFast + CoTA | 1.4 | 5.6 |
| **(a) Different dMLLMs.** | | | | **(b) Different cache methods.** | | |

Table 4: Comparison with additional caching methods.

| Method | ARR↓ | SRR↓ | | Method | ARR↓ | SRR↓ |
|---|---|---|---|---|---|---|
| LLaDA-V | 0.7 | 6.0 | | MMaDA | 0.7 | 6.0 |
| +Prefix-KV | 0.6 | 5.0 | | +Prefix-KV | 0.7 | 6.0 |
| +D$^3$ToM | 1.2 | 8.0 | | LaViDa | 0 | 0 |
| +FastdLLM | 0.7 | 6.0 | | +Prefix-KV | 0 | 0 |

### 5.2 RESULTS ON REPEAT CURSE EVALUATION

Table 2 presents the effectiveness of our method in mitigating the repeat curse on a VQA captioning task using 500 randomly sampled images from COCO Lin et al. (2014). SRR measures the likelihood of repetition occurring, while ARR evaluates its severity. We report the mean values of both metrics across samples. The experiments reveal a catastrophic Repeat Curse after applying caching techniques, with long-text responses being particularly prone to repetition. The results demonstrate that our method, CoTA, along with its two key components, CTAE and CTEV, effectively mitigates the repetition issues introduced by caching. Specifically, under the long-response setting, CoTA reduces ARR and SRR by 96% and 92%, respectively; under the short-response setting, it reduces ARR and SRR by 85% and 95%, respectively. Furthermore, we report the Repeat Curse results on MMaDA. As shown in Table 3(a), our method exhibits strong generalization ability, achieving improvements of 86% and 45% on ARR and SRR, respectively.

To evaluate the generalizability of CoTA across different caching methods, we further test its repetition-mitigation performance on SlowFast. As shown in Table 3(b), the CoTA, along with its components CTAE and CTEV, demonstrates consistent effectiveness. However, we find that some caching methods, such as D$^3$ToM, Fast-dLLM, and Prefix KV Cache, do not introduce the Repeat Curse, as shown in Table 4. By comparing these methods, we identify a common characteristic: they almost not cache the states of output tokens. Prefix KV Cache periodically caches prefix tokens (the prompt part) while recomputing the states of output tokens at every step. This setting corresponds to the experimental configuration in Table 1(a) and is consistent with the conclusions drawn in Section 3.2. D$^3$ToM builds upon Prefix KV Cache by introducing visual-token merging, which also applies only to the prompt portion. In contrast, Fast-dLLM recomputes the states of all output tokens after the block's starting position at every decoding step.

### 5.3 GENERALIZATION AND EFFICIENC STUDY

We report the generalization and efficiency analysis of CoTA in Table 5. Our method demonstrates consistent performance improvements and repetition mitigation across open-domain natural and mathematical tasks. When the baseline LLaDA-V employs caching techniques, it achieves substantial efficiency gains; however, it also exhibits the Repeat Curse and a drop in accuracy on both

Table 5: Generalization and Efficiency Analysis of CoTA.

| Method | LLaVA$^w$ | | | | MathVista | | | |
|---|---|---|---|---|---|---|---|---|
| | Score↑ | ARR↓ | TPS↑ | FLOPs↓ | Score↑ | ARR↓ | TPS↑ | FLOPs↓ |
| LLaDA-V | 70.1 | 1.3 | 7.3 | 16.1 | 59.7 | 0 | 8.3 | 14.1 |
| Ours | 70.5 | 1.1 | 5.1 | 17.8 | 59.8 | 0 | 7.5 | 15.3 |
| LLaDA-V w/ dLLM-Cache | 63.2 | 7.3 | 23.1 | 3.2 | 54.9 | 6.9 | 31.0 | 3.9 |
| Ours w/ dLLM-Cache | 69.9 | 1.4 | 20.3 | 5.1 | 59.7 | 1.3 | 28.7 | 5.0 |

benchmarks. In contrast, our method effectively overcomes this limitation of caching, yielding improvements of 11% in Score and 80% in ARR on LLaVA$^W$, and 9% in Score and 81% in ARR on MathVista. For efficiency evaluation, our method introduces only a small and acceptable computational overhead: on LLaVA$^w$, Tokens Per Second (TPS) decreases by 2.8, while Floating Point Operations (FLOPs) increase by 1.9.

## 5.4 ABLATION AND QUALITATIVE STUDY

Table 6 presents the ablation study of the hyperparameters involved in CoTA, including the weighting factor $\alpha$ in Equation 13, the minimum gain $\gamma_{\min}$ in Equation 7, and the number of context tokens. CoTA achieves its best performance when $\alpha = 0.75$, $\gamma_{\min} = 0.5$, and the number of context tokens is set to 3, yielding an ARR of 1.2% and an ACC of 23.1. We also report more detailed ablation analyses of CTEV and CTAE in Appendix E.4. Figure 9 illustrates the impact of CTAE on the model's intrinsic information-flow

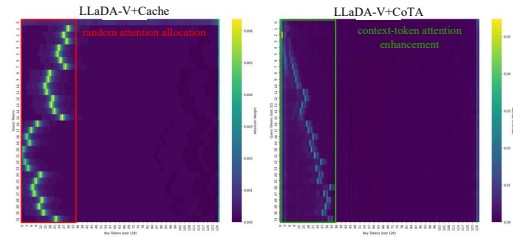

Figure 9: The intervention effect of CTAE.

pattern. After applying caching, the model exhibits a random attention distribution, whereas CoTA restores the "context tokens as anchors" pattern by strengthening attention toward contextual tokens.

Table 6: Ablation results on hyperparameters evaluated on the MathVerse benchmark.

| $\gamma_{\min}$ | $\alpha$ | ARR↓ | ACC↑ | $\gamma_{\min}$ | $\alpha$ | ARR↓ | ACC↑ | $\gamma_{\min}$ | $\alpha$ | ARR↓ | ACC↑ |
|---|---|---|---|---|---|---|---|---|---|---|---|
| | 0.25 | 2.8 | 21.4 | | 0.25 | 2.1 | 22.1 | | 0.25 | 3.5 | 19.6 |
| 1 | 0.5 | 2.2 | 21.9 | 1 | 0.5 | 2.0 | 22.5 | 1 | 0.5 | 3.8 | 19.2 |
| | 0.75 | 2.3 | 22.1 | | 0.75 | 1.7 | 22.3 | | 0.75 | 3.1 | 19.2 |
| | 0.25 | 2.9 | 20.0 | | 0.25 | 1.6 | 22.2 | | 0.25 | 3.9 | 19.0 |
| 0.5 | 0.5 | 2.5 | 20.2 | 0.5 | 0.5 | 1.4 | 22.0 | 0.5 | 0.5 | 3.4 | 18.2 |
| | 0.75 | 2.1 | 20.1 | | 0.75 | 1.2 | 23.1 | | 0.75 | 3.3 | 19.1 |
| (a) Context tokens number = 1 | | | | (b) Context tokens number = 3 | | | | (c) Context tokens number = 5 | | | |

## 6 CONCLUSION AND LIMITATIONS

From an information-flow perspective, this work reveals the underlying reasons why caching techniques in dMLLMs lead to repeated token generation, known as the Repeat Curse: (1) caching output tokens disrupts the intrinsic 'context tokens as anchors' information-flow pattern of dMLLMs, resulting in randomized attention distributions; and (2) neighboring context tokens exhibit persistently high deep-layer entropy during decoding, causing the model to generate uncertain and repeated outputs. Building on these insights, we propose CoTA, a plug-and-play approach to mitigate the repetition. Due to the limited availability of baseline dMLLM architectures and caching methods, CoTA has not yet been validated on a broader range of models and caching strategies.

ACKNOWLEDGMENTS

This work was conducted at the Fujian Key Laboratory of Pattern Recognition and Image Understanding, Xiamen University of Technology. It was supported by the National Natural Science Foundation of China (Grant No. 62576301) and the Major Science and Technology Plan Project on Future Industry Fields of Xiamen City (Grant No. 3502Z20241027).

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

## A    RELATED WORKS

**Diffusion Language Models.** Motivated by the remarkable success of diffusion models in image generation Ho et al. (2020); Rombach et al. (2022); Podell et al. (2024); Esser et al. (2024); Sohl-Dickstein et al. (2015); Song et al. (2021), recent studies have introduced the diffusion process into language modeling, including both continuous diffusion models Lovelace et al. (2023); Li et al. (2022); ZHANG et al. (2025); Xue et al. (2024); Lin et al. (2023a); Gong et al. (2023); Han et al. (2023); Strudel et al. (2022); Dieleman et al. (2022); Chen et al. (2023); Karimi Mahabadi et al. (2024); Graves et al. (2025); Lin et al. (2023b); Gulrajani & Hashimoto (2023) and discrete diffusion models Rout et al. (2025); Campbell et al. (2022); Zheng et al. (2025); Gat et al. (2024); Ye et al. (2023); Zheng et al. (2023); Austin et al. (2021); Hoogeboom et al. (2021); He et al. (2023); Meng et al. (2022); Shi et al. (2024); Ou et al. (2025); Nie et al. (2025a); Gong et al. (2025). The latest masked diffusion models Nie et al. (2025b); Zhu et al. (2025a); Ye et al. (2025a); Gong et al. (2025); He et al. (2025); Liu et al. (2025a) have successfully scaled dLLMs to 8B parameters and demonstrated performance comparable to autoregressive large language models Touvron et al. (2023a;b); Devlin et al. (2019); Grattafiori et al. (2024); DeepSeek-AI et al. (2024); Yang et al. (2024a); Gunasekar et al. (2023); Chiang et al. (2023).

**Multimodal Large Language Models (MLLMs).** Building upon the unprecedented generative capabilities of large language models (LLMs)Touvron et al. (2023a;b); Devlin et al. (2019); Grattafiori et al. (2024); DeepSeek-AI et al. (2024); Yang et al. (2024a); Gunasekar et al. (2023); Chiang et al. (2023), MLLMs have emerged as powerful systems that extend LLM architectures to process over multimodal inputs. Probabilistic modeling approaches for MLLMs generally fall into three paradigms: (i) autoregressiveLiu et al. (2023b;a); Bai et al. (2025; 2024); Zhu et al. (2025c); Chen et al. (2024b); Achiam et al. (2023); Li et al. (2025a); Guo et al. (2025); Zhang et al. (2024a); Wei et al. (2025b); Zhang et al. (2026); Yip et al. (2026b); Zhao et al. (2025b;a); Yip et al. (2026a), (ii) autoregressive–diffusion hybrid Bao et al. (2023); Xie et al. (2025a); Zhou et al. (2025); Ma et al. (2025b); Yu et al. (2025), and (iii) pure diffusion Swerdlow et al. (2025); Li et al. (2025d). Building on the recent breakthroughs in dLLMs, the latest dMLLMs You et al. (2025); Yang et al. (2025b); Li et al. (2025b) exploit the language modeling capabilities of dLLMs, coupled with effective training pipelines, to deliver performance on par with leading autoregressive and hybrid counterparts.

**Diffusion-based Multimodal Large Language Models (dMLLMs).** The latest diffusion-based large language models (dLLMs) Nie et al. (2025b); Zhu et al. (2025a); Ye et al. (2025a); Gong et al. (2025); He et al. (2025); Liu et al. (2025a); Yang et al. (2025c); Wen et al. (2025); Xie et al. (2025b); Zhu et al. (2025b); Tian et al. (2025); Dieleman et al. (2022) have been successfully scaled to 8B parameters, achieving performance comparable to state-of-the-art autoregressive large language models Touvron et al. (2023a;b); Devlin et al. (2019); Grattafiori et al. (2024); DeepSeek-AI et al. (2024); Yang et al. (2024a); Gunasekar et al. (2023); Chiang et al. (2023). Furthermore, by integrating visual instruction tuning and architectural extensions, dLLMs have recently been extended to dMLLMs Yang et al. (2025b); Li et al. (2025b); You et al. (2025); Yu et al. (2025), demonstrating promising multimodal capabilities.

**Caching Methods of dLLMs and dMLLMs.** Although state-of-the-art dLLMs Nie et al. (2025b); Zhu et al. (2025a); Ye et al. (2025a) and dMLLMs Yang et al. (2025b); Li et al. (2025b); You et al. (2025); Yu et al. (2025) demonstrate promising performance, their substantial inference latency remains a key limitation for practical deployment. Owing to their intrinsic bidirectional attention mechanism, these models cannot directly adopt the KV caching strategies Pope et al. (2023) used in autoregressive models. Several effective caching methods have been proposed to accommodate the characteristics of bidirectional attention. Specifically, **Prefix KV Cache** Li et al. (2025b) transfers the KV caching mechanism from autoregressive models to dLLMs by periodically caching prompt tokens, while recomputing the states of output tokens at every decoding step. **dLLM-Cache** Liu et al. (2025b) introduces a V-vary caching strategy. It observes that prompt features exhibit high similarity over long sampling intervals and therefore applies a longer caching interval to prompt tokens, reusing their states to avoid redundant computation. In contrast, output tokens remain stable only over shorter sampling intervals and are thus cached with a shorter interval. Moreover, based on the similarity of output tokens across adjacent steps, a certain proportion of tokens with low similarity is selectively recomputed. **SlowFast** Wei et al. (2025a) builds upon dLLM-Cache by introducing a dynamic sampling strategy that divides decoding into a cautious, exploratory decoding phase followed by a fast, deterministic parallel decoding phase. **MaskKV** Huang et al. (2025)

further improves upon dLLM-Cache by addressing the memory overhead introduced by caching, incorporating a fine-grained cache eviction strategy on top of dLLM-Cache. **D³ToM** Chang et al. (2025) extends Prefix KV Cache by introducing adaptive visual-token merging to alleviate token redundancy. **Fast-dLLM** Wu et al. (2025) proposes a caching strategy based on the block-wise decoding process of dLLMs: at each step, it recomputes the states of all tokens after the start of the current block, while caching the states of tokens preceding it. Other recent works include Ma et al. (2025a); Hu et al. (2025); Song et al. (2025b); Wang et al. (2025a).

**Repeated Token Generation.** Early research on text repetition and degeneration in model predictions primarily focused on improving sampling strategies for language models Holtzman et al. (2020), such as nucleus sampling and top-k sampling. Several studies treat repetition control as a training objective and introduce various training-time strategies to mitigate repetitive generation Welleck et al. (2020); Lagutin et al. (2021); Xu et al. (2022). Other methods focus on incorporating repetition-aware penalties during the sampling stage Zhu et al. (2023).

In recent years, research on repetitive text generation has expanded from conventional language generation models to modern LLMs. Yang et al. (2024b) estimates the likelihood of future repetition using an n-gram LM constructed from the generated prefix and imposes penalties based on this estimate. Ginart et al. (2025) introduces a repetition penalty from a compression-based perspective, while Li et al. (2023b) highlights how repeated tokens in training corpora can exacerbate repetition during inference. Some recent work further investigates the underlying mechanisms behind repetition. DUC Yao et al. (2025) employs Sparse Autoencoders (SAEs) Cunningham et al. (2023) to identify latent features that become highly activated when a model produces repeated tokens, referred to as repetition features. They attribute repetition to the activation of these features at specific layers and mitigate repetition by suppressing them. Wang et al. (2024) employs model-editing techniques to locate FFN neurons strongly associated with repetition, as well as neurons strongly related to the main task. By intersecting these neuron sets, they filter out neurons purely tied to repetition and perform targeted edits.

Distinct from the above approaches, our work examines repetition from an information-flow perspective. By comparing the information-flow patterns that emerge during repetitive generation with those observed under normal conditions, we identify the underlying mechanisms that lead to repetition. Our analysis shows that in dMLLMs, context tokens act as anchors that aggregate and propagate information. However, the introduction of cache disrupts this aggregation pattern and consequently triggers repetitive outputs. In addition, our layer-wise entropy analysis reveals that context tokens involved in local repetition exhibit a failure of entropy convergence in deeper layers. It is important to note that existing research on repetitive outputs in MLLMs and dMLLMs remains limited. Prior studies on MLLMs discuss repetition primarily in the context of hallucination and demonstrate that certain decoding strategies can reduce this effect Tong et al. (2025a); Huang et al. (2023). Recent dMLLM work has reported the presence of repetition but has not conducted deeper investigation into its causes Yu et al. (2025).

## B  ADDITIONAL INFORMATION-FLOW VISUALIZATIONS AND ANALYSES

We present additional information-flow visualizations to further substantiate our key finding that context tokens act as anchors. As shown in Figure 10, LLaDA-V exhibits a clear "right-diagonal" attention pattern, indicating that context tokens in close relative positions aggregate information and absorb attention across layers and decoding steps. A similar phenomenon is observed in MMaDA, as shown in Figure 11. Moreover, as discussed earlier, introducing caching techniques significantly disrupts the information-flow pattern of LLaDA-V. We further confirm this effect on MMaDA (as shown in Figure 12), where locally guided, context-driven attention under standard decoding degenerates into randomized attention distributions when caching is applied.

## C  RANDOM REMASKING

As discussed in Section 3.1, although dMLLMs can predict masked tokens at all positions in each decoding step, the model progressively refines its predictions by remasking a subset of tokens and re-predicting them in subsequent denoising steps. We have previously introduced the low-confidence remasking strategy. Random remasking is another token-state transition sampling strategy. Specif-

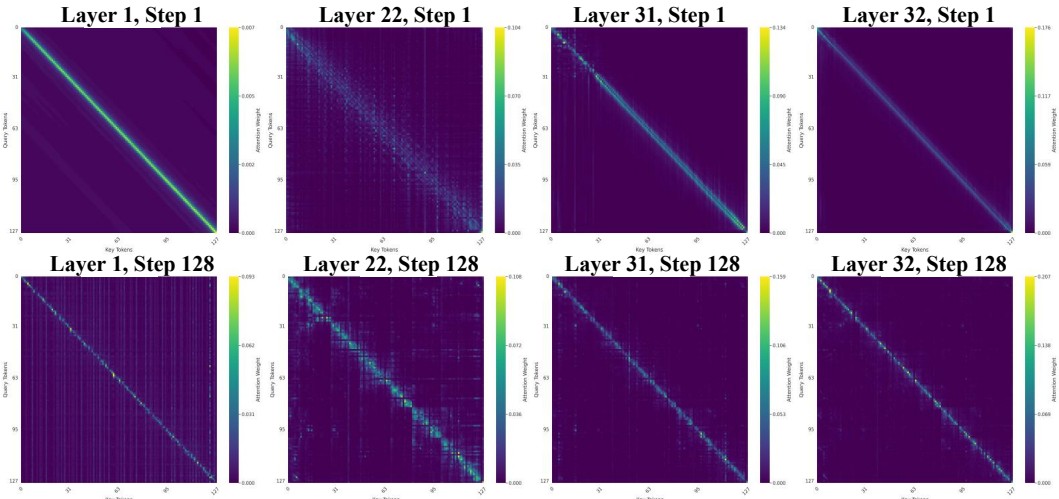

Figure 10: Information-flow visualization of LLaDA-V.

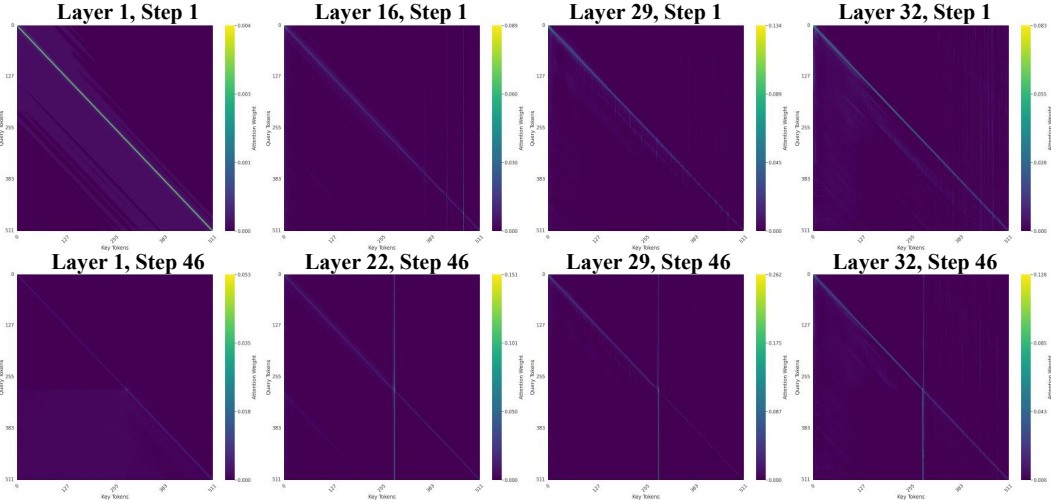

Figure 11: Information-flow visualization of MMaDA.

ically, for each token state $\mathcal{S}^t_{(i)}$ at decoding step $t$, if $\mathcal{S}^t_{(i)} \neq$ [MASK], the state remains unchanged, i.e., $\mathcal{S}^{t-1}_{(i)} = \mathcal{S}^t(i)$. If $\mathcal{S}^t_{(i)} =$ [MASK], then $\mathcal{S}^t_{(i)}$ is set to $\hat{\mathcal{S}}^t_{(i)}$ with probability $1 - (k-1)/k$, and remains masked with probability $(k-1)/k$.

## D MAXIMUM REPETITION LENGTH, AVERAGE REPETITION LENGTH, AND 95TH-PERCENTILE REPETITION LENGTH

Given an input sequence of length $M$, $\{\mathcal{T}_i\}_{i=1}^M$, we record the length of each consecutive identical segment $r_k$ as follows:

$$r_k = |\{\, \mathcal{T}_i \mid \mathcal{T}_i = \mathcal{T}_{i+1} = \cdots = \mathcal{T}_{i+r_k-1} \,\}|, \quad k = 1, 2, \ldots, K, \tag{14}$$

where $K$ denotes the total number of segments. Furthermore, we obtain all token repetition segments $rep\_runs$ as follows:

$$rep\_runs = \{\, r_k \mid r_k \geq 2, \ k = 1, 2, \ldots, K \,\}. \tag{15}$$

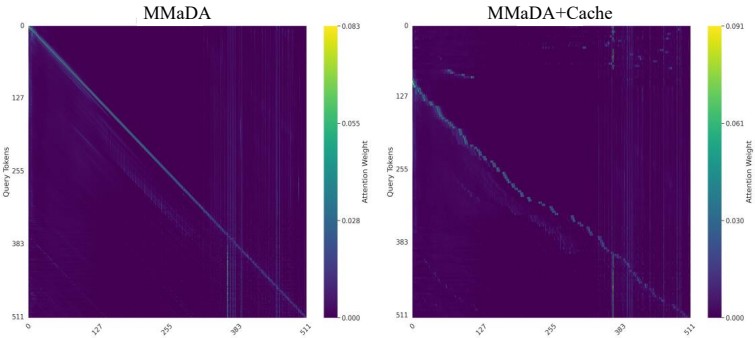

Figure 12: Information-flow visualization of MMaDA.

| Model | Type | LLM Tower | DocVQA | ChartQA | MMStar | $MME^p$ | $Seed^I$ | MMBench |
|---|---|---|---|---|---|---|---|---|
| LLaVA1.5 | AR | Vicuna-7B | - | - | - | 1510 | 66.1 | 64.3 |
| Qwen2-VL | AR | Qwen2-7B | - | 83.0 | 60.7 | - | - | - |
| DeepSeek-VL | AR | DeepSeek-7B | - | - | - | - | 70.4 | 73.2 |
| LLaVA-OV | AR | Qwen2-7B | - | 80.0 | 61.7 | 1580 | 75.4 | 80.8 |
| MetaMorph | AR+Diff. | LLaMA3.1-8B | - | 37.1 | - | - | 71.8 | 75.2 |
| JanusFlow | AR+Diff. | DeepSeek-1.3B | - | 64.6 | - | 1333 | 70.5 | 74.9 |
| LLaMA3-V | Diff. | LLaMA3-8B | 86.2 | 77.8 | 56.5 | 1581 | 76.6 | 79.8 |
| LLaDA-V | Diff. | LLaDA-8B | 83.9 | 78.3 | 60.1 | 1507 | 74.8 | 82.9 |
| LLaDA-V w/dllm-cache | Diff. | LLaDA-8B | 82.1 | 78.1 | 58.3 | 1410 | 72.1 | 83.0 |
| Ours | Diff. | LLaDA-8B | 84.1 | 78.4 | 59.3 | 1523 | 73.9 | 83.1 |

Table 7: Benchmark results of different MLLMs on multiple multimodal evaluation datasets.

Based on rep_runs, the maximum repetition length (MRL), average repetition length (ARL), and 95th-percentile repetition length (95pRL) can be obtained from Equations 16, 17, and 18, respectively.

$$\text{MRL} = \max(\text{rep\_runs}). \tag{16}$$

$$\text{ARL} = \frac{1}{|\text{rep\_runs}|} \sum_{r \in \text{rep\_runs}} r. \tag{17}$$

$$\text{95pRL} = \text{Quantile}_{0.95}(\text{rep\_runs}). \tag{18}$$

# E   ADDITIONAL EXPERIMENTAL ANALYSIS

## E.1   COMPARISON WITH OTHER MLLMS

Table 7 presents a performance comparison with other paradigm MLLMs. Our goal is not absolute performance maximization, but rather to mitigate the output degradation of baseline models when using cache. Results across six benchmarks demonstrate the effectiveness of our method.

## E.2   ANALYSIS OF CACHE MECHANISM AND REPEAT CURSE

In this section, we conduct a more detailed analysis of the repeat curse induced by caching, examining how different components and design choices in caching techniques relate to repetition. We begin by providing a supplementary analysis of the experimental results in Table 1:

(a) We begin by examining the prompt token recomputation interval, while fixing the output token recomputation interval to 1 and setting the similarity threshold to 0. As shown in Table 1(a), periodic caching of prompt tokens has negligible impact on repetition.

---

**Algorithm 2** Context Tokens Attention Enhancement

---

**Require:** Attention weights $A \in \mathbb{R}^{L \times H \times T \times T}$, temperature $\tau > 0$, lower bound $\gamma_{\min} \in (0, 1]$
**Ensure:** Enhanced attention $\tilde{A} \in \mathbb{R}^{L \times H \times T \times T}$
 1:  $\tilde{A} \leftarrow A$
 2: **for** $\ell = 1$ to $L$ **do**
 3:    *(layer index)*
 4:    **for** $h = 1$ to $H$ **do**
 5:      *(head index)*
 6:      $D_{i,j} \leftarrow |i - j| \quad \forall i, j \in \{1, \ldots, T\}$
 7:      $g_{i,j} \leftarrow \exp\left(-(D_{i,j}/\tau)^2\right)$
 8:      $\mathcal{G}_{i,j} \leftarrow \gamma_{\min} + (1 - \gamma_{\min}) \cdot g_{i,j}$
 9:      $\tilde{A}_{\ell,h} \leftarrow A_{\ell,h} \odot \mathcal{G}$
10:    **end for**
11: **end for**
12: **return** $\tilde{A}$

---

(b) We then fix the prompt token recomputation interval to 25 and set the similarity threshold to 0, varying only the output token recomputation interval. As reported in Table 1(b), we observe that periodic caching of output tokens induces repetition, and longer recomputation intervals lead to higher repetition rates.

(c) In the dLLM-Cache framework, a subset of output tokens is adaptively recomputed at each step based on a similarity threshold. To analyze this further, we fix the prompt token recomputation interval to 25 and the output token recomputation interval to 7, while varying the similarity threshold. As shown in Table 1(c), the threshold plays a critical role: lower thresholds correspond to higher repetition rates.

(d) Finally, we compare dLLM-Cache with prefix KV cache Kwon et al. (2023); Li et al. (2025b) to analyze the effect of different reuse policies. We fix the prompt token recomputation interval to 25, the output token recomputation interval to 7, and the similarity threshold to 0.25. As shown in Table 1(d), prefix KV cache, which only reuses cached states for prefix tokens, does not trigger repetition.

These four experiments reveal that the prompt recomputation interval has little effect on repetition behavior, while the output token recomputation interval, similarity threshold, and caching policy exert a substantial influence. In general, the fewer output tokens that are recomputed at each decoding step, the more likely the model is to fall into repetition.

We hypothesize that this is because the token states within the prompt sequence remain relatively static, so caching them has minimal impact on future predictions. In contrast, output tokens evolve across decoding steps. Caching these dynamic tokens forces the model to rely on outdated states, reducing uncertainty during prediction—ultimately contributing to the non-convergence of context-token entropy in deeper layers.

### E.3   Information Flow Across Different dMLLMs

As illustrated in Figure 11, we visualize the attention maps of MMaDA to analyze the flow of information between tokens. We observe a similar pattern to LLaDA-V, where context tokens function as anchors, continually attracting attention across layers. Interestingly, we also detect vertical bands of concentrated attention in Layer 22 at Step 46, resembling the "attention sink" Huang et al. (2023) phenomenon commonly seen in autoregressive MLLMs. However, after introducing the cache, the original attention pattern of MMaDA is also disrupted (as shown in Figure 12).

### E.4   Additional Analyses of CoTA

We first present the workflow of CTAE in Algorithm 2. CTAE applies a distance-based decay term to attention values, thereby enhancing attention toward context tokens. CTEV further introduces deep-layer context-token entropy as a penalty to the voting scores. Together, these two components jointly alleviate the repeat curse. N-gram penalties Holtzman et al. (2020), as a simple and widely

| Methods | ACC↑ | ARR↓ |
|---|---|---|
| LLaDA-V + cache | 54.9 | 6.9 |
| n-gram ($n = 2$) | 54.0 | 2.4 |
| n-gram ($n = 3$) | 54.3 | 3.2 |
| CoTA | **59.7** | **1.3** |

(a) Comparison between CoTA and n-gram penalties.

| Methods | ARR↓ | SRR↓ |
|---|---|---|
| Decay Matrix | **1.8**% | **8**% |
| AliBi Bias Matrix | 3.9% | 10% |

(b) Comparison of different attention biasing methods.

Table 8: Experimental results of CoTA and related techniques.

used decoding-time technique, are commonly applied to control repetition. We further compare the performance of CoTA and n-gram penalties in Table 8. We find that although n-gram penalties can reduce repetition, their effectiveness is inferior to that of CoTA. Moreover, applying n-gram penalties often leads to a degradation in model accuracy, likely due to forced token substitution, which disrupts semantic coherence and lowers output quality. In addition, we replace the decay matrix used in CTAE's attention biasing design with an Alibi bias penalty matrix Tang et al. (2025b)to evaluate its effect. As shown in Table 8, the results confirm that using the Decay Matrix for attention intervention in CTAE is more effective.

| Methods | 16 | 32 | 64 | 128 |
|---|---|---|---|---|
| LLaDA-V | 2% | 1% | 1% | 1% |
| LLaDA-V+cache | 88% | 85% | 79% | 75% |
| LLaDA-V+CoTA | 17% | 14% | 10% | 8% |

Table 9: Repeat curse comparison under different block lengths. The evaluation metric is the sample repetition rate.

We further analyze the sensitivity of CTEV through additional experiments. As shown in Table 9, we evaluate the CoTA performance under different block lengths during decoding. We observe that our method consistently reduces the repetition rate across all tested block configurations, demonstrating its robustness to different window sizes.

| Layer range | 512 | | 128 | |
|---|---|---|---|---|
| | ARR↓ | SRR↓ | ARR↓ | SRR↓ |
| LLaDA-V+cache | 13.9% | 85% | 9.3% | 75% |
| 1–10 | 12.1% | 81% | 8.4% | 73% |
| 11–20 | 12.4% | 83% | 9.1% | 74% |
| 21–25 | 11.3% | 73% | 7.9% | 65% |
| 25–30 | 5.6% | 23% | 4.3% | 19% |
| 26–30 | 2.9% | 10% | 1.8% | 8% |
| 26–31 | 3.2% | 12% | 2.1% | 10% |
| 26–32 | 3.3% | 12% | 2.1% | 9% |

Table 10: Performance comparison under different layer ranges.

We further present results across different layer ranges in Table 10. We find that computing the cumulative entropy using layers 26–30 yields the best performance. We speculate that this is because the entropy values in this depth range exhibit the largest separation between normal and abnormal modes, while incorporating additional layers introduces unnecessary noise, leading to diminished performance.

| | LLaDA-V+cache | | |
|---|---|---|---|
| **Repeated word** | **the** | **of** | **a** |
| Repetition ratio | 98% | 76% | 56% |

Table 11: Statistics of repeated words generated by LLaDA-V with cache.

### E.5 LINGUISTIC ATTRIBUTE ANALYSIS OF REPEATED TOKENS

In this section, we analyze the linguistic attributes of repeated tokens by computing their frequency statistics over 200 samples that exhibit repetition. As shown in Table 11, the top-3 most frequently repeated words are "the," "of," and "a." We observe that repeated tokens are predominantly function words carrying low semantic content, which is also consistent with the case study presented in Figure 1. This indicates a common tendency of the model when generating uncertain or repetitive text.

| Method | throughput (tok/s) ↑ | latency (s/sample) ↓ |
|---|---|---|
| LLaDA-V | 1.8 | 17.4 |
| +prefix KV cache | 2.7 | 13.3 |
| +dllm-cache | 4.9 | 6.1 |
| +dllm-cache+CTEV | 4.1 | 7.2 |
| +dllm-cache+CTAE | 4.6 | 6.4 |
| +dllm-cache+CoTA | 3.8 | 7.7 |
| MMaDA | 0.5 | 32.1 |
| +dllm-cache | 2.0 | 19.6 |
| +dllm-cache+CTEV | 1.5 | 22.0 |
| +dllm-cache+CTAE | 1.7 | 21.4 |
| +dllm-cache+CoTA | 1.3 | 24.0 |

Table 12: Throughput and latency comparison on DocVQA.

### E.6 MORE EFFICIENCY ANALYSIS

We further perform a comprehensive efficiency analysis on DocVQA, covering prefix-KV, dLLM-cache, CoTA, and its two components (CTAE and CTEV). As shown in Table 12, integrating CoTA introduces only a modest and acceptable increase in latency, along with a slight reduction in throughput, yet it still delivers substantial improvements over the baseline. Moreover, to provide a more holistic assessment of efficiency, we additionally report latency and throughput results on both LLaDA-V and MMaDA.

| Evaluation data | DocVQA | ChartQA | MMStar | MME | SEED |
|---|---|---|---|---|---|
| Max generation length | 32 | 16 | 2 | 2 | 2 |
| Block length | 32 | 16 | 2 | 2 | 2 |
| Decode steps | 16 | 16 | 2 | 2 | 2 |
| Batchsize | 1 | 1 | 1 | 1 | 1 |

Table 13: Evaluation configuration details of different datasets.

## F CONFIGURATION DETAILS

### F.1 EVALUATION CONFIGURATION DETAILS OF DIFFERENT DATASETS

We report detailed information about the evaluation setup in Table 13, including the maximum generation length, block length, decoding steps, and batch size for different benchmarks.

### F.2 ARCHITECTURE OF LLaDA-V

Following the baseline dMLLM LLaDA-V You et al. (2025), the language tower adopts LLaDA-8B-Instruct Nie et al. (2025b), the vision tower employs siglip2-so400m-patch14-384 Tschannen et al. (2025), and the projector is a two-layer MLP.

## G DECLARATION OF THE USE OF GENERATIVE AI (LARGE LANGUAGE MODELS)

Generative AI tools, including Grammarly and ChatGPT, were used solely for grammar checking and language polishing. All technical content, experimental design, data analysis, and conclusions were generated and verified exclusively by the human authors. The use of AI tools does not affect the originality or authorship of this work.

## H ETHICS STATEMENT

This work seeks to uncover the underlying causes of the Repeat Curse in dMLLMs with cache from an information-flow perspective and to propose mitigation strategies that enhance the output performance of diffusion-based multimodal large language models (dMLLMs). The proposed methods, CoTA, is developed using only publicly available dMLLMs (LLaDA-V), caching approaches (dLLM-Cache), and benchmark datasets (e.g., MME, MMBench, SEED), without relying on any private, sensitive, or human-subject data. These techniques do not introduce biases beyond those inherent to the base models, and are designed to supplement rather than replace human supervision in critical applications. Nevertheless, while our methods can effectively mitigate response repetition, they cannot guarantee the elimination of errors or misleading outputs. Hence, caution is advised in practical deployments.

## I REPRODUCIBILITY STATEMENT

To ensure full reproducibility, we provide the following resources: (1) Code: The complete implementation of CoTA, including Context Tokens Attention Enhancement (CTAE) and Context Tokens Entropy-Guided Voting (CTEV), will be publicly released on GitHub upon publication. (2) Hyperparameters: For dLLM-Cache, all experiments fix the hyperparameters at $\alpha = 25\%$, $\mathcal{E}_p = 25$, and $\mathcal{E}_s = 7$. The parameter details for CoTA are described in Section 5.4, while those for the evaluation setup are provided in Section F. (3) Evaluation: We use standard, publicly available benchmarks (e.g., MME, MMBench, SEED) along with their official evaluation scripts. (4) Compute: Experiments are conducted on NVIDIA A800 80GB GPUs. Inference latency is reported using TPS and FLOPs (Table 5). (5) Models and Cache Methods: We evaluate open-source dMLLMs, specifically LLaDA-V (8B). The cache method is based on the official implementation of dLLM-Cache for LLaDA-V. No proprietary data or models are used in this work.

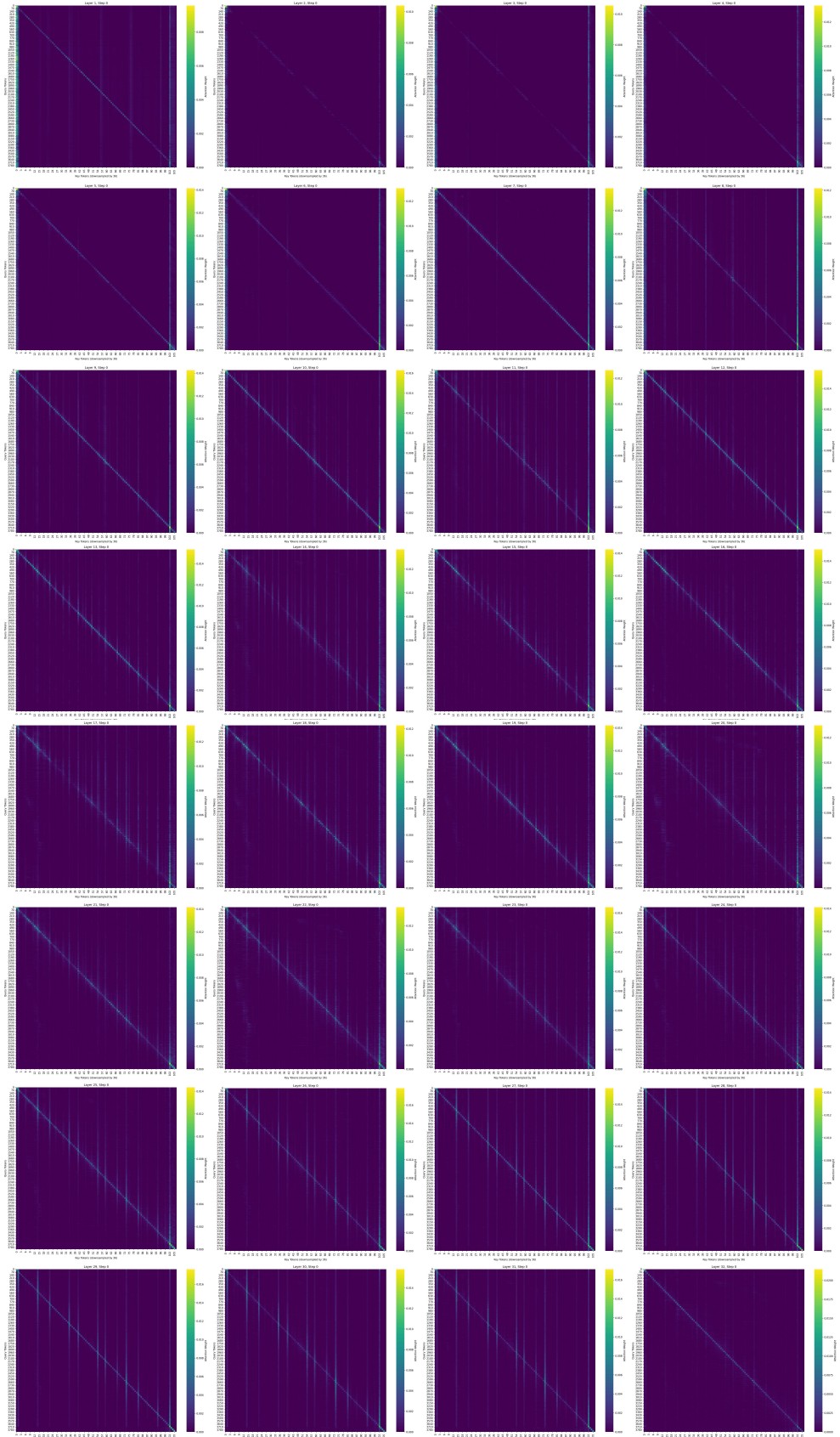

Figure 13: Full Attention Maps of Each Layer of LLaDA-V.

