# OpenReview forum: "Context Tokens are Anchors: Understanding the Repeat Curse in dMLLMs from an Information Flow Perspective"
_ICLR.cc/2026/Conference — ICLR 2026 Poster_

### Official Review · Reviewer_bFZ4 · 2025-10-16

**Soundness:** 3
**Presentation:** 4
**Contribution:** 4
**Rating:** 8
**Confidence:** 4

**Summary:**

The paper successfully identifies and names the phenomenon of cache-induced repetition in dMLLMs during inference, referred to as the “Repeat Curse,” demonstrating a strong awareness of the problem setting.
By examining information entropy and attention flow, the paper reveals the mechanism behind repetitive generation, showcasing solid theoretical explanatory power. The paper proposes the insight that “Context Tokens are Anchors,” uncovering how context tokens act as anchors across layers and stabilize the decoding process. The proposed CoTA module (CTAE + CTEV) is a plug-and-play solution that requires no additional training, making it highly practical for real-world deployment. The work not only validates the mitigation effect on repetition but also demonstrates strong generalization across different tasks and compatibility with existing cache strategies.

**Strengths:**

From the full workflow of "Problem Definition → Mechanism Analysis → Method Design → Experimental Validation → Detail Supplementation", this work features a complete structure, rigorous logic, clear innovation, and strong practicality. Its core contributions—revealing the repetition mechanism of dMLLMs + caching and proposing CoTA, a lightweight deduplication method—hold both academic and application value.

**Weaknesses:**

1. First of all, this should not be the first time that the “repeat curse” problem has been raised. It would be better to include some citations and add an analysis of previous methods that address the repeat curse in the Related Work section. In particular, please highlight solutions proposed for LLMs and MLLMs. This can help readers understand that your approach is not just a direct adaptation from LLMs to dMLLMs, but rather an innovation tailored specifically to the structure of dMLLMs.
2. You could also consider adding another attention map after applying CoTA as a supplement.

**Questions:**

The types of repetition I've seen roughly include local word repetition, sentence pattern looping, and semantic looping. This article seems to only observe local word repetition. Is it because dLLMs only have this type of repetition?

---

> ### Author Response · Authors · 2025-11-20
> **Response to bFZ4**
>
> ### **Response to bFZ4:**
> We sincerely thank you for taking the time to review our manuscript. We appreciate your positive evaluation of our work as well as the insightful comments you provided. Below, we respond to each of the identified weaknesses and questions in detail.
>
> ### **1. Weakness 1**
> **Comment:**
> *First of all, this should not be the first time that the “repeat curse” problem has been raised. It would be better to include some citations and add an analysis of previous methods that address the repeat curse in the Related Work section. In particular, please highlight solutions proposed for LLMs and MLLMs. This can help readers understand that your approach is not just a direct adaptation from LLMs to dMLLMs, but rather an innovation tailored specifically to the structure of dMLLMs.*
>
> **Response to Weakness 1:**
> Thank you very much for your suggestion. In the revised manuscript, we have added citations and discussion of prior work on the “repeat curse” in the Related Work section (as shown in Appendix G of the updated manuscript). We also provide a brief overview of methods designed to mitigate repetition in LLMs and MLLMs, in order to better highlight the motivation and uniqueness of our approach.
>
> ### **2. Weakness 2**
> **Comment:**
> *You could also consider adding another attention map after applying CoTA as a supplement.*
>
> **Response to Weakness 2:**
> Visualizing the attention maps after introducing CoTA is indeed important for understanding the effect of our method. In the updated manuscript, we have added these visualizations (see Figure 9) and included a detailed discussion in Appendix D. We observe that models using cache tend to exhibit randomly distributed attention, while applying CoTA strengthens attention on context tokens. Your suggestion has significantly improved the quality of the paper.
>
> ### **3. Question 1**
> **Comment:**
> *The types of repetition I've seen roughly include local word repetition, sentence pattern looping, and semantic looping. This article seems to only observe local word repetition. Is it because dLLMs only have this type of repetition?*
>
> **Response to Question 1:**
> Our discussion of repetition in dLLMs is presented only as a case study in the Appendix K. Our intention is to highlight that dLLMs may also exhibit a form of the repeat curse that deserves further investigation, and to encourage the community to explore this phenomenon in future work. However, the specific repetition pattern shown in the case study is incidental, and the repetition behaviors in dLLMs are certainly not limited to local word-level repetition. In our recent exploration of the long-context capabilities of dLLMs such as LLaDA and Dream, we observed multiple types of repetition directly in long-context scenarios, including both local word repetition and cyclic sentence-level patterns. These observations were made using LLaDA-8B-Base and LLaDA-8B-Instruct on benchmarks such as LongBench and Needle-In-A-Haystack.

---

> > ### Comment · Reviewer_bFZ4 · 2025-11-25
> >
> > Thank you for your work. You have addressed all my concerns. I will keep my original scores.
> >
> > BTW, It seems that some appendices are not referred to in the main text, I recommend you add them on in the camera-ready version.

---

> ### Author Response · Authors · 2025-11-25
>
> We are delighted to hear that your concerns have been resolved, and we are very grateful for your consistently positive evaluation of our work. We will revise the camera-ready version to ensure that all appendix materials are properly integrated and clearly cited. Thank you again for your valuable comments and suggestions.

---

### Official Review · Reviewer_XZLN · 2025-11-01

**Soundness:** 2
**Presentation:** 2
**Contribution:** 2
**Rating:** 4
**Confidence:** 3

**Summary:**

This paper studies repetition in diffusion-based MLLMs with cache (“Repeat Curse”) and analyzes it via information flow. The authors observe that (i) context tokens serve as “anchors” that aggregate information and guide predictions; (ii) under normal decoding, the entropy of context tokens decreases and converges across deeper layers; (iii) with cache enabled, attention becomes randomized and context-token entropy fails to converge, which correlates with repetition. Building on these observations, they propose CoTA, consisting of Context-Token Attention Enhancement (CTAE) and Context-Token Entropy-guided Voting (CTEV). Experiments on several multimodal benchmarks and a 500-image captioning evaluation show reduced repetition and preserved task performance when using cache.

**Strengths:**

1. Clear empirical characterization of repetition with interpretable metrics and visualizations.
2. A training-free, plug-in style method (CoTA) that targets the identified failure mode and is simple to implement.
3. Results suggest repetition can be mitigated without large performance drops.

**Weaknesses:**

1. Mechanism not adequately explained.
The paper does not convincingly explain why cache introduction leads to non-convergence of context-token entropy in deeper layers. Current evidence is largely correlational (entropy trajectories + repetition metrics) without ablations that isolate which cache component (e.g., recomputation interval, similarity thresholds, reuse policy) drives the phenomenon. A causal link is missing.
2. Lack of generality (single model + single cache).
All main claims—the “Repeat Curse,” the information-flow analysis, and CoTA efficacy—are validated on a single model (LLaDA-V 8B) and a single cache method (dLLM-Cache). This undermines the generality of the conclusions. Although the conclusion section acknowledges this, the paper still reads as over-claimed relative to its evidence base.
3. CTEV design choices under-justified.
CTEV aggregates cumulative entropy from layers 26–30. The choice appears ad-hoc and may be tightly coupled to a 32-layer backbone. No sensitivity analysis (different windows, learned weights, different depths) is reported, so robustness remains unclear.
4. Presentation/editing issues.
Page 2, paragraph 2: figure references are incorrect/mixed, making the early narrative confusing.

**Questions:**

Please refer to the weakness part.

---

> ### Author Response · Authors · 2025-11-20
> **Response to XZLN (1/3)**
>
> ### **Response to XZLN:**
> Thanks a lot for the time and effort you invested in providing the detailed reviews. Your review comments have greatly improved the completeness of the work and the quality of the manuscript. Below are our responses to the weaknesses you pointed out.
>
> ### **1. Weakness 1**
> **Comment:**
> *Mechanism not adequately explained. The paper does not convincingly explain why cache introduction leads to non-convergence of context-token entropy in deeper layers. Current evidence is largely correlational (entropy trajectories + repetition metrics) without ablations that isolate which cache component (e.g., recomputation interval, similarity thresholds, reuse policy) drives the phenomenon. A causal link is missing.*
>
> **Response to Weakness 1:**
> Your comment is very insightful. We have added additional studies on LLaDA-V+cache to further ablate the cache component. The experimental results and detailed analyses are presented in Table 5 and Appendix A of the updated manuscript.
>
> (a) Prompt Recomputation Interval:
> We first conduct an ablation analysis of the prompt-token recomputation interval under the dLLM-Cache setting. In this experiment, the output-token re-computation interval is fixed to 1, and the similarity thresholds are set to 0.
> |Prompt Token Recomposition Interval |1|5|15|25|
> |-|-|-|-|-|
> |SRR↓ |0|0|0|0|
>
> Conclusion 1: Periodic caching of prompt tokens almost never leads to repetition.
>
> (b) Output Recomputation Interval:
> We fix the prompt-token recomputation interval to 25 and set the similarity thresholds to 0, and then perform an ablation analysis on the output-token recomputation interval.
> |Output Token Recomposition Interval|1|3|5|7|
> |-|-|-|-|-|
> |SRR↓|0|79.9|87.4|89.7|
>
> Conclusion 2: Periodic caching of output tokens leads to repetition, and the repetition rate increases as the re-computation interval becomes longer.
>
> (c) Similarity thresholds：
> We fix the prompt-token recomputation interval to 25 and the output-token recomputation interval to 7, and perform an ablation study on the similarity thresholds.
> |Similarity Thresholds|0|0.25|0.5|0.75|1|
> |-|-|-|-|-|-|
> |SRR↓|89.7|75.0|69.8|29.7|0|
>
> Conclusion 3: The similarity threshold influences the severity of the repeat curse—lower thresholds lead to higher repetition rates.
>
> (d) Reuse Policy:
> We fix the prompt-token recomputation interval of dLLM-Cache to 25, the output-token recomputation interval to 7, and the similarity threshold to 0.25. We then use prefix KV cache as a comparison method and perform an ablation study on the reuse policy.
> |Reuse Policy| prefix KV cache | dllms-cache |
> |-|-|-|
> |SRR↓|0|75.0|
>
> Conclusion 4: The prefix KV cache reuses only the historical states of prefix tokens and therefore almost never causes repetition.
>
> Summary: Among all cache components, the re-computation interval for prompt tokens almost never leads to the repeat curse. In contrast, the re-computation interval for output tokens, the similarity threshold, and the reuse policy significantly influence the severity of repetition. Overall, generating fewer newly updated output tokens per step makes the model more prone to repetition. We hypothesize that this is because the token states in the prompt sequence remain relatively static, so caching them has minimal impact on generation. In contrast, the token states in the output sequence evolve dynamically across decoding steps; caching them causes the model to rely on outdated token states for prediction, reducing the uncertainty during generation. This, in turn, is a key factor contributing to the non-convergence of context-token entropy in deeper layers.

---

> ### Author Response · Authors · 2025-11-20
> **Response to XZLN (2/3)**
>
> ### **2. Weakness 2**
> **Comment:**
> *Lack of generality (single model + single cache). All main claims—the “Repeat Curse,” the information-flow analysis, and CoTA efficacy—are validated on a single model (LLaDA-V 8B) and a single cache method (dLLM-Cache). This undermines the generality of the conclusions. Although the conclusion section acknowledges this, the paper still reads as over-claimed relative to its evidence base.*
>
> **Response to Weakness 2:**
> We fully agree that generalization experiments are crucial for this work. We have therefore added additional comparisons across more models and cache methods.
>
> (a) More Models:
>
> We have added the following experiments on MMaDA:
> |Method| ARR ↓ | SRR↓ |
> |-|-|-|
> |MMaDA|0.7|6.0|
> |+dllm-cache|4.4| 55.0|
> |+dllm-cache+CTAE|2.4|29.0|
> |+dllm-cache+CTEV|0.8|35.0|
> |+dllm-cache+CoTA|0.6|30.0|
>
> Repeat Curse Analysis: After applying dLLMs-Cache, MMaDA also exhibits repetitive behavior, demonstrating that the repeat curse exists across different dMLLMs. The experimental results and detailed analyses are presented in Table 6 and Appendix B.1 of the updated manuscript.
>
> Information-Flow Analysis: As shown in Figures 6 and 7 of the uploaded manuscript, we have added information-flow visualizations for MMaDA. MMaDA exhibits a similar information-flow pattern in which context tokens serve as anchors that absorb attention across layers. Introducing cache disrupts this pattern.
>
> CoTA Efficacy Analysis: The results for CoTA and its components (CTAE and CTEV) further demonstrate the generalization ability of our method. The experimental results and detailed analyses are presented in Table 6 and Appendix B.3 of the updated manuscript.
>
> (b) More Cache Methods:
>
> We further evaluated the repetition behavior when applying the prefix-KV cache method (referenced from https://docs.vllm.ai and LaViDa, NeurIPS 2025) on LLaDA-V and LaViDa. The results are as follows:
> |Model |ARR↓|SRR↓|
> |-|-|-|
> |LLaDA-V|0|0|
> |+prefix KV cache|0|0|
> |LaViDa|0|0|
> |+prefix KV cache|0|0|
>
> We find that applying the prefix KV cache does not induce repetition in the model. This is likely because the prefix KV cache only reuses the states of prefix tokens (prompt tokens and image tokens) and does not involve caching suffix tokens (output text tokens). This observation is consistent with the conclusion in our Response to Weakness 1, where we show that repetition is primarily associated with caching output tokens. (Additionally, it is worth noting that the official implementation of dLLM-Cache currently does not support LaViDa.) The experimental results and detailed analyses are presented in Table 6 and Appendix B.4 of the updated manuscript.

---

> ### Author Response · Authors · 2025-11-20
> **Response to XZLN (3/3)**
>
> ### **3. Weakness 3**
> **Comment:**
> *CTEV design choices under-justified. CTEV aggregates cumulative entropy from layers 26–30. The choice appears ad-hoc and may be tightly coupled to a 32-layer backbone. No sensitivity analysis (different windows, learned weights, different depths) is reported, so robustness remains unclear.*
>
> **Response to Weakness 3:**
> Thank you for your suggestion. We have added more details on the method design as well as additional discussions on sensitivity.
>
> (a) Design Details of CTEV:
>
> The selection of layers 26–30 in CTEV is derived from both multi-sample statistics and single-sample analysis. We compute the entropy of each layer (out of 32 attention layers) over 100 samples, and then sum and average the entropy values of the same layer across different samples. The entropy distribution across layers is shown in Figure 8(a) of the uploaded manuscript. By comparing the entropy curves before and after applying the cache, we observe that the introduction of caching causes significant changes in entropy specifically in layers 26–30 (as analyzed in Section 3.3 of the paper), which directly motivates the design of CTEV. In addition, we further analyze the sum of entropy values across layers 26–30 for different models. As shown in Figure 8(b) of the uploaded manuscript, applying cache consistently leads to poor entropy convergence in deeper layers, which further validates the rationale for selecting layers 26–30 in CTEV. The experimental results and detailed analyses are presented in Figure 8 and Appendix C of the updated manuscript.
>
> (b) Sensitivity Analysis of CTEV:
>
> (i) Following your suggestion, we analyze different window settings by testing various block lengths (with the maximum text length set to 128) and report the resulting repetition curse behavior in terms of the Sample Repetition Rate.
> |Block Length|16|32|64|128|
> |-|-|-|-|-|
> |LLaDA-V|2%|2%|1%|1%|
> |LLaDA-V+cache|88%|85%|79%|75%|
> |LLaDA-V+CoTA|17%|14%|10%|8%|
>
> We find that our method consistently reduces the repetition rate across different block lengths, demonstrating its stability.
>
> (ii) Since our method is training-free, it does not involve any learnable weights. In addition, LLaDA-V has not yet released official training scripts or implementation details, and considering the potential instability in reproduction, we have not attempted training at this stage. We plan to investigate learnable weights once the official training scripts for LLaDA-V become available.
>
> (iii) Following your suggestion, we have added an analysis of CTEV with different depths, and the results are as follows:
> |Layer range |ARR ↓ (512)|SRR ↓(512) |ARR ↓ (128)|SRR ↓(128)|
> |-|-|-|-|-|
> |LLaDA-V+cache |13.9%|85%|9.3%|75%|
> |1–10|12.1%|81%|8.4%|73%|
> |11–20|12.4%|83%|9.1%|74%|
> |21–25|11.3%|73%|7.9%|65%|
> |25–30|5.6%|23%|4.3%|19%|
> |26–30|2.9%|10%|1.8%|8%|
> |26–31|3.2%|12%|2.1%|10%|
> |26–32|3.3%|12%|2.1%|9%|
>
> We find that computing the CTEV entropy-accumulation penalty using layers 26–30 yields the best final results. We hypothesize that this is because the entropy in this range exhibits the highest separability between normal and abnormal patterns, whereas incorporating additional layers introduces unnecessary noise.
>
> The experimental results and detailed analyses are presented in Table 7, Table 8, and Appendix C of the updated manuscript.
>
> ### **4. Weakness 4**
> **Comment:**
> *Presentation/editing issues. Page 2, paragraph 2: figure references are incorrect/mixed, making the early narrative confusing.*
>
> **Response to Weakness 4:**
> Thank you for pointing out the incorrect figure references. We have corrected them in the newly uploaded manuscript.

---

> ### Author Response · Authors · 2025-11-25
> **Inquiry Regarding Rebuttal Feedback**
>
> Dear Reviewer XZLN,
>
> We sincerely appreciate your time and effort in reviewing our manuscript and offering valuable suggestions. We provided detailed clarifications in response to your questions a few days ago. If you have any additional feedback, concerns, or questions regarding our response, we would greatly appreciate hearing from you and welcome further discussion.

---

> > ### Comment · Reviewer_XZLN · 2025-11-28
> > **Reviewer XZLN — Response to Authors**
> >
> > Thanks for the careful rebuttal and the added experiments. The new ablations identify output-token caching as the main driver of repetition, while prefix-KV does not exhibit the issue; the added model and the layer-window sensitivity help justify the CTEV design. These results address the majority of my original concerns.
> >
> > My remaining, minor concern is about applicability and reporting clarity. Since prefix-KV appears safe, it would help readers decide when CoTA is the right choice if you could report some results like latency/throughput for prefix-KV, output-caching (dLLM-cache), and dLLM-cache + CoTA, and also provide component-level overhead (CTAE vs CTEV). I saw the efficiency discussion in section 5.4 (TPS/FLOPs), but it may not be sufficient to guide adoption decisions; there also seem to be small inconsistencies/typos (FPS vs TPS usage and the statement about FLOPs reduction). Clarifying these points would make the practical trade-offs clearer.
> >
> > With these notes, the rebuttal resolves the bulk of my concerns, and I am willing to raise my score.

---

> > > ### Author Response · Authors · 2025-11-28
> > > **Response to XZLN**
> > >
> > > ### **Response to XZLN:**
> > >
> > > We are glad to hear that our previous clarifications have addressed your earlier concerns. Regarding the remaining questions about efficiency, we provide detailed latency and throughput results, along with component-level analyses, as reported below.
> > >
> > > Following your suggestions, we conducted a thorough efficiency analysis on the DocVQA benchmark, covering prefix-KV, dLLM-cache, CoTA, and the two CoTA components (CTAE and CTEV). As shown in the table below, introducing CoTA incurs a small and acceptable increase in latency and decrease in throughput, while still providing substantial improvements over the baseline. We have also corrected the wording and typographical issues you pointed out.
> > >
> > > | Method                | throughput (tok/s) ↑ | latency (s/sample) ↓ |
> > > |-----------------------|----------------------|------------------------|
> > > | LLaDA-V               | 1.8                  | 17.4                   |
> > > | +prefix KV cache      | 2.7                  | 13.3                   |
> > > | +dllm-cache           | 4.9                  | 6.1                    |
> > > | +dllm-cache+CTEV      | 4.1                  | 7.2                    |
> > > | +dllm-cache+CTAE      | 4.6                  | 6.4                    |
> > > | +dllm-cache+CoTA      | 3.8                  | 7.7                    |
> > > | MMaDA                 |    0.5               |    32.1                |
> > > | +dllm-cache           |     2.0              |        19.6             |
> > > | +dllm-cache+CTEV      |       1.5            |          22.0           |
> > > | +dllm-cache+CTAE      |       1.7            |          21.4           |
> > > | +dllm-cache+CoTA      |      1.3             |          24.0           |
> > >
> > > The detailed experimental results and analyses are provided in Table 11 and Appendix G of the updated manuscript. Thank you once again for your insightful comments and for your positive assessment of our work.

---

### Official Review · Reviewer_bzXt · 2025-11-01

**Soundness:** 2
**Presentation:** 1
**Contribution:** 1
**Rating:** 2
**Confidence:** 4

**Summary:**

This paper studies the repetition curse that arises when caching is applied to accelerate inference in multimodal diffusion large language models (dMLLMs). While caching improves efficiency, it leads to repetitive text generation. The authors propose **CoTA (Context Tokens are Anchors)** to mitigate this issue by preserving attention coefficients of context tokens (typically 3 neighboring tokens) and downweighting attention for keys that are far from the current position. Context tokens are heuristically chosen based on proximity to the query position. Experiments on 500 randomly sampled images from the COCO dataset for image captioning demonstrate that CoTA alleviates repetition while maintaining generated caption quality.

**Strengths:**

This paper makes an interesting observation that enabling caching in multimodal diffusion large language models can distort attention coefficients, leading to degraded generation quality.
To address this, the authors propose an attention reweighting mechanism that enhances the attention coefficients of a few key context tokens (typically 3), termed **context anchors**, to preserve semantic consistency while maintaining the efficiency benefits of caching.

**Weaknesses:**

1. **Line 82:** The text refers to Figure 2.c, but there is no subfigure (c) in Figure 2.

2. **Generalization:** Is the proposed approach generic, or is it specific to LLaDA? Can it also be applied to other diffusion large-language models?

3. **Line 139:** The authors mention prior studies showing the existence of certain anchor tokens in autoregressive models. However, the related works section does not discuss anchors in diffusion language models, such as ADLM (https://openreview.net/pdf?id=E8adS5srds). It is important to include related work on anchoring in diffusion models since this paper primarily focuses on context tokens as anchors in diffusion language models, which was first proposed in ADLM.

4. **Figure 3:** The font is too small and difficult to read.

5. **Line 280 Typo:** “Equation 5 6” → “Equations 5 and 6.”

6. **Figure 4 Incorrect labeling:** Subfigure “(a), (a)” → “(a), (b).”

7. **Applicability of Attention Enhancement:** The main idea of attention enhancement is relevant only when caching changes the original attention pattern. Recent models such as BD3LM can better handle caching, so this problem may not arise. Can the authors verify that their observations hold for such models?

8. **Equation 7:** Why is the entropy normalized by $\log(V)$?

9. **Ablation Study (Table 4):** Increasing the number of context tokens from 3 to 5 drastically reduces performance, which seems counterintuitive and contradicts the principle of attention mechanisms that capture long-range dependencies. Could the authors explain this behavior?

**Questions:**

Please see the weaknesses above.

---

> ### Author Response · Authors · 2025-11-20
> **Response to bzXt (1/3)**
>
> ### **Response to bzXt:**
> Thank you very much for your valuable comments. Below, we provide detailed responses to the concerns you raised and clarify any points that may have led to misunderstanding.
>
> ### **1. Weakness 1**
> **Comment:**
> *Line 82: The text refers to Figure 2.c, but there is no subfigure (c) in Figure 2.*
>
> **Response to Weakness 1:**
> Thank you for pointing out the incorrect citation in Figure 2. We have corrected this mistake in the updated manuscript and have also reviewed the entire paper to prevent similar issues.
> ### **2. Weakness 2**
> **Comment:**
> *Generalization: Is the proposed approach generic, or is it specific to LLaDA? Can it also be applied to other diffusion large-language models?*
>
> **Response to Weakness 2:**
> We fully understand your concerns regarding the generalization ability of our method. Therefore, in addition to LLaDA-V, we further evaluate the generalization of CoTA on another dMLLM, MMaDA. The generalization results on MMaDA are as follows:
> |Method|Adjacent Repetition Rate (ARR) ↓|Sample Repetition Rate (SRR) ↓|
> |-|-|-|
> |MMaDA|0.7|6.0|
> |+dllm-cache|4.4|55.0|
> |+dllm-cache+CTAE|2.4|29.0|
> |+dllm-cache+CTEV|0.8|35.0|
> |+dllm-cache+CoTA|0.6|30.0|
>
> We find that using dLLMs-Cache on MMaDA also triggers the repeat curse, demonstrating the generality of this issue. We further evaluate CoTA and its two key components (CTAE and CTEV) on MMaDA, and the results confirm the generalization ability of our method. The experimental results and detailed analyses are presented in Table 6, Appendix B.1, and Appendix B.3 of the updated manuscript.
>
> In addition, we would like to clarify that our baseline model is LLaDA-V rather than LLaDA, and that our study focuses on diffusion Multimodal Large Language Models (dMLLMs), not diffusion Large Language Models (dLLMs). That said, we plan to further investigate the Repetition Curse and the performance of CoTA on dLLMs and other unified models in future work.
> ### **3. Weakness 3**
> **Comment:**
> *Line 139: The authors mention prior studies showing the existence of certain anchor tokens in autoregressive models. However, the related works section does not discuss anchors in diffusion language models, such as ADLM (https://openreview.net/pdf?id=E8adS5srds). It is important to include related work on anchoring in diffusion models since this paper primarily focuses on context tokens as anchors in diffusion language models, which was first proposed in ADLM.*
>
> **Response to Weakness 3:**
> We fully agree that prior work on anchor tokens in diffusion language models, especially ADLM, is highly relevant to our study. Accordingly, we have added a discussion and citation in the Related Work section (Section 2) of the updated manuscript.
>
> In addition, we would like to clarify the essential difference between the notion of “anchors” in ADLM and the “anchors” referred to in our paper:
>
> (a) Different definitions: In ADLM, anchors are key tokens in the language sequence determined by semantic importance (such as low-frequency words or semantic keywords). In our work, anchors refer to a phenomenon discovered through information-flow visualization and internal mechanism analysis of the model.
>
> (b) Different functions: Anchors in ADLM are introduced to be observed earlier in long-context generation, thereby reducing semantic entropy at other positions. In our paper, anchors act as information aggregators that gather cross-layer information during self-attention computation.
>
> (c) Different domains: ADLM belongs to diffusion Language Models, typically with relatively small parameter scales (≤ 350M) and purely textual inputs. Our focus is on diffusion Multimodal Large Language Models (dMLLMs), which are significantly larger (8B) and operate in multimodal settings.
>
> (d) Different motivations: ADLM leverages anchors during training to encourage the model to decode anchor tokens earlier and improve performance. In contrast, our notion of anchors emerges from an information-flow analysis aimed at explaining the internal mechanisms of dMLLMs. This understanding allows us to propose a training-free intervention method by examining the relationship between the repeat curse and information-flow patterns.
>
> Of course, we greatly appreciate your comments regarding diffusion Language Models, which have strongly inspired our future work. We have added a brief discussion of diffusion Language Models (dLLMs) in Appendix K, and we plan to further investigate information flow, repetition phenomena, and related behaviors in dLLMs in future research. Some preliminary experimental analysis can already be found in our response to Reviewer bFZ4’s Question 1.

---

> ### Author Response · Authors · 2025-11-20
> **Response to bzXt (2/3)**
>
> ### **4. Weakness 4**
> **Comment:**
> *Figure 3: The font is too small and difficult to read.*
>
> **Response to Weakness 4:**
> Thank you for pointing this out. In the updated manuscript, we have adjusted the font size and image resolution of Figure 3 to improve clarity. We sincerely apologize for any inconvenience this may have caused during your reading.
> ### **5. Weakness 5**
> **Comment:**
> *Line 280 Typo: “Equation 5 6” → “Equations 5 and 6.”*
>
> **Response to Weakness 5:**
> Thank you very much for your comment. We have corrected the wording in the updated manuscript.
> ### **6. Weakness 6**
> **Comment:**
> *Figure 4 Incorrect labeling: Subfigure “(a), (a)” → “(a), (b).”*
>
> **Response to Weakness 6:**
> Thank you for pointing out the labeling error in the figure. We again apologize for any confusion caused by these formatting issues during your reading.
> ### **7. Weakness 7**
> **Comment:**
> *Applicability of Attention Enhancement: The main idea of attention enhancement is relevant only when caching changes the original attention pattern. Recent models such as BD3LM can better handle caching, so this problem may not arise. Can the authors verify that their observations hold for such models?*
>
> **Response to Weakness 7:**
> We understand your concerns and believe that there may be some misunderstandings regarding our work. We clarify these points as follows:
>
> (a) As noted by Reviewer qq7Y, dMLLMs and their associated cache mechanisms constitute an emerging research area, and the repeat curse is a critical and practically relevant issue. The most widely used dMLLMs include LLaDA-V and MMaDA. We conduct experiments on them to examine repetition behaviors under different cache methods and to evaluate the generalization of CoTA. The results reported for LLaDA-V in the paper, along with our additional results for MMaDA, are as follows:
> |Model|ARR↓|SRR↓|
> |-|-|-|
> |MMaDA|0.7|6.0|
> |+dllms-cache|4.4|55.0|
> |+dllms-cache+CoTA|0.6|30.0|
> |LLaDA-V|0.2|6.9|
> |+dllms-cache|14.3|82.3|
> |+dllms-cache+CoTA|1.2|6.3|
>
> (b) Due to architectural differences, existing cache mechanisms for dLLMs (e.g., Dream, LLaDA) and dMLLMs (e.g., LLaDA-V, MMaDA) are currently not compatible with BD3LM. Likewise, adapting the BD3LM caching strategy to dLLMs or dMLLMs is technically challenging. Research on cache designs for dLLMs and dMLLMs is still in its early stages. We will continue to follow newly released and compatible cache methods and incorporate them into future investigations.
>
> (c) Our study of the repeat curse on dMLLMs is conducted on multimodal benchmarks (such as VQA), whereas BD3LM is a language-only model. The substantial differences in both model type and evaluation benchmarks make it difficult to ensure a fair and controlled comparison. Therefore, we believe the statement “Recent models such as BD3LM can better handle caching, so this problem may not arise.” is not sufficiently rigorous.
>
> (d) As noted by Reviewer bFZ4, our work follows a complete and logically consistent workflow, from problem formulation to mechanism analysis, method design, experimental validation, and supplementary diagnostic studies. Any investigation on BD3LM would likewise need to adhere to this same rigorous process. In particular, studying BD3LM would require more extensive experimental design and information-flow analyses. This is indeed an interesting direction, and we plan to further explore these aspects in future extensions of our work.
> ### **8. Weakness 8**
> **Comment:**
> *Equation 7: Why is the entropy normalized by Log(V)?*
>
> **Response to Weakness 8:**
> We compute the cumulative deep-layer entropy and use it as a penalty term for the confidence score during decoding. Since the confidence score involves a softmax operation, we normalize the entropy to ensure that its value distribution is comparable to the confidence score and to improve numerical stability.

---

> ### Author Response · Authors · 2025-11-20
> **Response to bzXt (3/3)**
>
> ### **9. Weakness 9**
> **Comment:**
> *Ablation Study (Table 4): Increasing the number of context tokens from 3 to 5 drastically reduces performance, which seems counterintuitive and contradicts the principle of attention mechanisms that capture long-range dependencies. Could the authors explain this behavior?*
>
> **Response to Weakness 9:**
>
> We understand your concerns and explain the rationale below:
>
> (a) Our context tokens are designed to capture the local neighborhood of repeated tokens, rather than to expand the attention scope. This is different from the global receptive field typically discussed in language-modeling Transformers.
>
> (b) This phenomenon does not contradict the theory that attention mechanisms can capture long-range dependencies. In dMLLMs, each decoding step already leverages bidirectional self-attention to model global dependencies. Through cross-layer information aggregation, context tokens (which are adjacent in relative position) absorb attention and act as anchor tokens.
>
> (c) The length of the context-token window directly affects entropy computation. Increasing the window size may introduce unnecessary noise, making it more difficult for the model to reliably distinguish uncertain tokens.

---

> ### Author Response · Authors · 2025-11-25
> **Inquiry Regarding Rebuttal Feedback**
>
> Dear Reviewer bzXt,
>
> We sincerely appreciate your time and effort in reviewing our manuscript and offering valuable suggestions. We provided detailed clarifications in response to your questions a few days ago. If you have any additional feedback, concerns, or questions regarding our response, we would greatly appreciate hearing from you and welcome further discussion.

---

### Official Review · Reviewer_qq7Y · 2025-11-03

**Soundness:** 3
**Presentation:** 3
**Contribution:** 3
**Rating:** 6
**Confidence:** 3

**Summary:**

This paper identifies the "Repeat Curse," a text repetition issue caused by caching in diffusion-based Multimodal Large Language Models (dMLLMs). The authors analyze this from an information flow perspective, finding that caching disrupts the role of "context tokens" as information anchors and prevents their entropy from converging in deeper layers. To solve this, they propose CoTA, a training-free method with two components: Context-token Attention Enhancement (CTAE) to restore information flow, and Context-token Entropy-guided Voting (CTEV) to penalize uncertain predictions during decoding. Experiments show CoTA effectively reduces repetition and restores model performance on various multimodal benchmarks.

**Strengths:**

1. Addresses a critical and practical issue (inference acceleration side-effects) in the emerging field of dMLLMs.
2. Provides a technically sound analysis of the problem's root cause using information flow and entropy, which is more insightful than a purely phenomenological study.
3. Proposes a novel, training-free, and plug-and-play method (CoTA) that is shown to be highly effective.
4. The experiments are well-designed, featuring custom metrics to quantify the problem, extensive benchmark validation, and thorough ablation studies.

**Weaknesses:**

1. The study's conclusions are drawn from a single dMLLM architecture (LLaDA-V 8B), making it unclear if the "Repeat Curse" is a universal issue.
2. The analysis is tied to a single caching method (dLLM-Cache). The findings may not hold for other acceleration strategies.
3. Defining "context tokens" as a fixed-size local window feels empirical. The paper lacks a deeper theoretical justification for why these specific tokens act as anchors.
4. The paper does not investigate whether "anchor" tokens or repeating tokens possess specific linguistic properties (e.g., function words vs. content words).
5. The experiments do not compare CoTA against classic, simpler methods for controlling repetition, such as n-gram penalties or blocking.
The CTAE component is conceptually similar to attention biasing methods. A discussion of these related techniques is missing.

**Questions:**

1. How do you expect your findings to generalize to other dMLLM architectures (e.g., JanusFlow) and different model sizes?
2. Would the "Repeat Curse" still occur with alternative caching strategies, perhaps those with more adaptive update policies?
3. Why a fixed local window for "context tokens"? Have you investigated if the "anchor" role is dynamic or could involve non-adjacent tokens?
4. Did your analysis reveal if certain types of words (e.g., common function words like "the") are more prone to exhibiting high entropy and causing repetition?
5. What was the rationale for choosing layers 26-30 for the entropy calculation in CTEV? How sensitive is the performance to this specific layer range?
6. Could you comment on how CoTA compares to simpler decoding-time baselines like n-gram blocking, particularly regarding the trade-off between repetition control and text quality?

---

> ### Author Response · Authors · 2025-11-20
> **Response to qq7Y (1/3)**
>
> ### **Response to qq7Y:**
> We sincerely appreciate the time and effort you dedicated to reviewing our manuscript, as well as your positive evaluation and insightful comments. Below, we provide point-by-point responses to the weaknesses and questions you raised:
> ### **1. Weakness 1**
> **Comment:**
> *The study's conclusions are drawn from a single dMLLM architecture (LLaDA-V 8B), making it unclear if the "Repeat Curse" is a universal issue.*
>
> **Response to Weakness 1:**
> We agree that the repeat curse should be analyzed on more dMLLMs. We have added additional repeat curse experiments on MMaDA as follows:
> |Model|Adjacent Repetition Rate (ARR)↓|Sample Repetition Rate (SRR)↓|
> |-|-|-|
> |MMaDA|0.7|6.0|
> |+dllm-cache|4.4|55.0|
> |+dllm-cache+CTAE|2.4|29.0|
> |+dllm-cache+CTEV|0.8|35.0|
> |+dllm-cache+CoTA|0.6|30.0|
>
> We observe that MMaDA also exhibits repetition after applying cache, which further demonstrates the generality of the “Repeat Curse.” In addition, CoTA and its key components (CTAE and CTEV) show strong generalization, as their effectiveness is further validated on MMaDA. The corresponding experiments and detailed analyses can be found in Table 6 and Appendix B of the updated manuscript.
> ### **2. Weakness 2**
> **Comment:**
> *The analysis is tied to a single caching method (dLLM-Cache). The findings may not hold for other acceleration strategies.*
>
> **Response to Weakness 2:**
> Following your suggestion, we additionally report the repetition behavior of dMLLMs when using prefix KV cache (referencing https://docs.vllm.ai and [LaViDa, NeurIPS 2025]).
> |Model| ARR↓| SRR↓|
> |-|-|-|
> |LLaDA-V|0|0|
> |+prefix KV cache|0|0|
> |+dllm-cache|9.3|75.0|
> |LaViDa|0|0|
> |+prefix KV cache|0|0|
>
> The results show that prefix KV cache does not induce repetition in either LLaDA-V or LaViDa. The key difference between prefix KV cache and dLLMs-Cache is that prefix KV cache does not cache output tokens. To further analyze how this design relates to the repeat curse, we conducted an ablation study on the key components of dLLMs-Cache, and the results are as follows:
>
> (a) Prompt Recomputation Interval:
> We first conduct an ablation analysis of the prompt-token recomputation interval under the dLLM-Cache setting. In this experiment, the output-token re-computation interval is fixed to 1, and the similarity thresholds are set to 0.
> |Prompt Token Recomposition Interval|1|5|15|25|
> |-|-|-|-|-|
> |SRR↓ |0|0|0|0|
>
> Conclusion 1: Periodic caching of prompt tokens almost never leads to repetition.
>
> (b) Output Recomputation Interval:
> We fix the prompt-token recomputation interval to 25 and set the similarity thresholds to 0, and then perform an ablation analysis on the output-token recomputation interval.
>
> |Output Token Recomposition Interval |1|3|5|7|
> |-|-|-|-|-|
> |SRR↓|0|79.9|87.4|89.7|
>
> Conclusion 2: Periodic caching of output tokens leads to repetition, and the repetition rate increases as the re-computation interval becomes longer.
>
> (c) Similarity thresholds：
> We fix the prompt-token recomputation interval to 25 and the output-token recomputation interval to 7, and perform an ablation study on the similarity thresholds.
>
> |Similarity Thresholds|0|0.25|0.5|0.75|1|
> |-|-|-|-|-|-|
> |SRR↓|89.7|75.0|69.8|29.7|0|
>
> Conclusion 3: The similarity threshold influences the severity of the repeat curse—lower thresholds lead to higher repetition rates.
>
> Summary: For the three key components of dLLM-Cache, we find that the re-computation interval for prompt tokens almost never induces the repeat curse. In contrast, the re-computation interval for output tokens and the similarity threshold have a substantial impact on the severity of repetition. Overall, the fewer output tokens that are recomputed at each decoding step, the more likely the model is to produce repetition. In comparison, prefix KV cache recomputes all output tokens at every step and only caches prompt tokens, which explains why it does not exhibit repetitive behavior.
>
> We hypothesize that this difference arises because the token states within the prompt sequence remain relatively static, so caching them has minimal effect on generation. In contrast, the token states in the output sequence evolve dynamically across decoding steps; caching these dynamic states forces the model to rely on outdated information, thereby reducing uncertainty during generation and increasing the likelihood of repetition. The corresponding experiments and detailed analyses can be found in Table 5, Table 6, Appendix A, and Appendix B.4 of the updated manuscript.

---

> ### Author Response · Authors · 2025-11-20
> **Response to qq7Y (2/3)**
>
> ### **3. Weakness 3**
> **Comment:**
> *Defining "context tokens" as a fixed-size local window feels empirical. The paper lacks a deeper theoretical justification for why these specific tokens act as anchors.*
>
> **Response to Weakness 3:**
> The design of context tokens in CTEV is closely tied to entropy computation. As discussed in Section 3 of the paper, we observe that adjacent context tokens containing repetition exhibit non-convergent deep-layer entropy, whereas non-repetitive contexts show a clear convergence trend. This phenomenon is validated through statistical analysis across multiple samples and models (as shown in Figure 8 and Appendix C of the updated manuscript). This observation directly motivates the design of CTEV: the convergence behavior of deep-layer entropy serves as a reliable signal for identifying context tokens associated with repetition.
>
> In addition, the effectiveness of setting the number of context tokens to 3 can be theoretically explained. The evaluation metric for repetition is based on whether adjacent tokens are identical (as shown by ARR in Equation 4 of the manuscript). A window size of 3 naturally covers the two nearest positions around the target token, which correspond to the most common repetition spans observed in dMLLMs. This configuration offers robustness across different lengths of repeated-token patterns while minimizing the influence of noisy or irrelevant distant tokens.
> ### **4. Weakness 4**
> **Comment:**
> *The paper does not investigate whether "anchor" tokens or repeating tokens possess specific linguistic properties (e.g., function words vs. content words).*
>
> **Response to Weakness 4:**
>
> We agree that analyzing the linguistic attributes of repeated tokens is necessary. We conducted frequency statistics on repeated tokens across 200 samples that exhibit repetition. In the updated manuscript, Appendix E and Table 9 present the corresponding statistics and analyses. The top-3 most frequently repeated words are reported as follows:
> ||LLaDA-V+Cache|||
> |-|-|-|-|
> |repeated words|the|of|a|
> |repetition ratio|98%|76%|56%|
>
> We find that most repeated tokens are function words carrying low semantic content (e.g., “the,” “of,” “a”), which reflects a common tendency of the model when generating uncertain or repetitive words.
> ### **5. Weakness 5**
> **Comment:**
> *The experiments do not compare CoTA against classic, simpler methods for controlling repetition, such as n-gram penalties or blocking. The CTAE component is conceptually similar to attention biasing methods. A discussion of these related techniques is missing.*
>
> **Response to Weakness 5:**
>
> Thank you for your suggestion.
>
> (1) We have added a discussion of n-gram penalties, and the experimental results are as follows:
> |Methods|MathVista ACC|MathVista ARR|
> |-|-|-|
> |LLaDA-V+cache|54.9|6.9|
> |n-gram (n=2)|54.0|2.4|
> |n-gram (n=3)|54.3|3.2|
> |CoTA|59.7|1.3|
>
> We find that although n-gram penalties can reduce repetition, their effectiveness is inferior to that of CoTA. Moreover, while decreasing repetition, n-gram penalties also lead to a drop in model accuracy, likely because forced token substitutions disrupt semantic coherence and degrade output quality. The corresponding experiments and detailed analyses can be found in Table 10.a and Appendix F of the updated manuscript.
>
> (2) We agree that it is necessary to discuss alternative attention biasing methods. We replaced the decay matrix in CTAE with an AliBI bias penalty matrix and obtained the following results:
> |Methods|ARR|SRR|
> |-|-|-|
> |CTAE with Decay Matrix|1.8%|8%|
> |CTAE with AlIBI Bias penalty Matrix|3.9%|10%|
>
> The experimental results demonstrate the effectiveness of using our decay matrix for attention intervention in CTAE. The corresponding experiments and detailed analyses can be found in Table 10.b and Appendix F of the updated manuscript.
> ### **6. Question 1**
> **Comment:**
> *How do you expect your findings to generalize to other dMLLM architectures (e.g., JanusFlow) and different model sizes?*
>
> **Response to Question 1:**
>
> CoTA is training-free and plug-and-play, giving it strong theoretical adaptability across different dMLLM architectures and model sizes. As shown in the additional experiments in our Response to Weakness 1, we further validate the generalization ability of CoTA on MMaDA. However, existing cache mechanisms for popular dMLLMs such as LLaDA-V, LaViDa, and MMaDA remain very limited. We plan to conduct more comprehensive comparisons across various models and scales once more cache-related implementations for dMLLMs become publicly available.

---

> ### Author Response · Authors · 2025-11-20
> **Response to qq7Y (3/3)**
>
> ### **7. Question 2**
> **Comment:**
> *Would the "Repeat Curse" still occur with alternative caching strategies, perhaps those with more adaptive update policies?*
>
> **Response to Question 2:**
>
> We hypothesize that the “Repeat Curse” can be effectively mitigated as more adaptive update policies are developed. As discussed in our response to Weakness 2, more adaptive mechanisms for handling output tokens are intuitively promising for alleviating repetition. The design of CoTA incorporates both attention-level intervention and entropy-based adjustment, and we hope these ideas may inspire future work in this direction.
> ### **8. Question 3**
> **Comment:**
> *Why a fixed local window for "context tokens"? Have you investigated if the "anchor" role is dynamic or could involve non-adjacent tokens?*
>
> **Response to Question 3:**
>
> As mentioned in our Response to Weakness 3, the purpose of introducing a local window for context tokens is to capture locally uncertain tokens, which helps identify potential repetitive context tokens. The idea of using dynamic or non-adjacent tokens is indeed interesting, and we plan to explore it further in future work. However, our current mechanism analysis (as shown in Figure 4 and Section 3 of the manuscript) does not yet provide sufficient support for dynamic or non-adjacent token selection.
> ### **9. Question 4**
> **Comment:**
> *Did your analysis reveal if certain types of words (e.g., common function words like "the") are more prone to exhibiting high entropy and causing repetition?*
>
> **Response to Question 4:**
>
> As noted in our Response to Weakness 4, we followed your suggestion and analyzed the semantic attributes of repeated tokens. Our analysis shows that low-semantic function words such as “of” and “the” constitute the majority of repeated tokens, indicating that the model tends to generate low-semantic function words when repetition occurs.
> ### **10. Question 5**
> **Comment:**
> *What was the rationale for choosing layers 26-30 for the entropy calculation in CTEV? How sensitive is the performance to this specific layer range?*
>
> **Response to Question 5:**
>
> Please refer to our Response to Weakness 3, as well as Appendix C and Figure 8 in the updated manuscript. Through both multi-sample statistics and single-sample analysis, we observe that the introduction of cache leads to significant entropy changes in Layers 26–30. This strong separation motivates the selection of Layers 26–30 for our method.
>
> In addition, following your suggestion, we have added a sensitivity analysis on the choice of layer ranges as follows:
> |Layer range |ARR ↓ (512)|SRR ↓(512) |ARR ↓ (128)|SRR ↓(128)|
> |-|-|-|-|-|
> |LLaDA-V+cache |13.9%|85%|9.3%|75%|
> |1–10|12.1%|81%|8.4%|73%|
> |11–20|12.4%|83%|9.1%|74%|
> |21–25|11.3%|73%|7.9%|65%|
> |25–30|5.6%|23%|4.3%|19%|
> |26–30|2.9%|10%|1.8%|8%|
> |26–31|3.2%|12%|2.1%|10%|
> |26–32|3.3%|12%|2.1%|9%|
>
> The experimental results show that selecting Layers 26–30 yields the best performance, and expanding the range to include additional neighboring layers does not provide further improvement. The results and detailed analysis are presented in Table 8 and Appendix C of the updated manuscript.
> ### **11. Question 6**
> **Comment:**
> *Could you comment on how CoTA compares to simpler decoding-time baselines like n-gram blocking, particularly regarding the trade-off between repetition control and text quality?*
>
> **Response to Question 6:**
>
> Including this discussion has enriched our work, and we appreciate your suggestion. N-gram blocking mitigates repetition by penalizing repeated token spans during decoding, but this straightforward technique does not investigate the internal mechanisms that cause repetition, nor does it address the limitations introduced by cache from a principled perspective. Moreover, as shown in our experiments, although n-gram blocking can reduce repetition, it often degrades output quality and leads to a drop in accuracy (ACC). For more detailed results and analysis, please refer to our Response to Weakness 5, as well as Table 10 and Appendix F in the updated manuscript.

---

> ### Author Response · Authors · 2025-11-25
> **Inquiry Regarding Rebuttal Feedback**
>
> Dear Reviewer qq7Y,
>
> We sincerely appreciate your time and effort in reviewing our manuscript and offering valuable suggestions. We provided detailed clarifications in response to your questions a few days ago. If you have any additional feedback, concerns, or questions regarding our response, we would greatly appreciate hearing from you and welcome further discussion.

---

### Comment · Area_Chair_uAtW · 2025-11-23
**Reviewer-Author Discussion**

Hi Reviewers,

Please kinly and actively participate in the review-author dicussion, raise your further concerns so that the authors can explain more, and make your final decisions.

---

### Author Response · Authors · 2025-11-30
**Summary of all rebuttal responses**

Dear Area Chairs/Program Chairs,

Thank you for overseeing the review of our submission. We sincerely appreciate the time and effort of all reviewers and the AC in managing this process. We have answered all the questions raised by the reviewers. Reviewers bFZ4 and XZLN indicated that their previous concerns have been resolved and expressed willingness to raise scores from **8642** to **8662**. Reviewer qq7Y maintains a positive assessment of our work (score **6**). Although reviewer bFZ4 did not participate in the rebuttal/discussion stage, we have responded comprehensively to all issues they raised. Below, we summarize the feedback from each reviewer:

### ✅ 1. Reviewer qq7Y/bzXt/XZLN: The generality of the conclusions and the method.

All three reviewers suggested adding experiments on different dMLLMs to support our claims regarding the "Repeat Curse," the information-flow analysis, and the effectiveness of CoTA.

- (1) We reported the Repeat Curse results, information-flow analysis, and the generalization performance of CoTA on MMaDA.

Reviewers qq7Y and XZLN suggested evaluating repetition under different caching methods.

- (2) We added experiments evaluating repetition behavior in LLaDA-V and LaViDa under prefix-KV cache.

Reviewer XZLN suggested adding ablation studies on cache components.

- (3) We added comprehensive ablation experiments on recomputation interval, similarity thresholds, and reuse policy.

Reviewer XZLN stated: "**These results address the majority of my original concerns.** My remaining, minor concern is ... and **I am willing to raise my score (4->6)**." (November 28, 16:03). This indicates that our additional experiments and analyses have effectively addressed their concerns. Although reviewer bzXt did not participate in the rebuttal, we believe that our rebuttal directly address their concerns regarding generalization.

### ✅ 2. Reviewer qq7Y/XZLN: More evaluation of CTEV.
Both reviewers noted that, in the design of CTEV, the use of accumulated entropy from layers 26–30 requires additional motivation experiments to demonstrate that this choice is neither ad-hoc nor purely empirical.

- (1) We provided both multi-sample statistics and single-sample analyses to illustrate the motivation behind the CTEV design. In addition, we conducted statistical experiments on different dMLLMs.

Both reviewers requested a discussion on the sensitivity of CTEV.

- (2) We added results using different window sizes to demonstrate the generalizability of the method.
- (3) We conducted detailed experiments and analysis across different layer depths.

### ✅ 3. Reviewer bzXt/XZLN: Citation, labeling, and typographical errors.
- We have corrected all citation, labeling, and typographical errors pointed out by the two reviewers.

### ✅ 4. Reviewer bFZ4: Provide additional visualization results.
- We added more attention visualizations of our method.

Reviewer bFZ4 say "**You have addressed all my concerns**. I will keep my original scores **(8)**." November 25nd 18:24 clock.

### ✅ 5. Reviewer XZLN: More efficiency analysis. (After rebuttal)
- We added detailed efficiency analyses in both the rebuttal and the updated manuscript.

The Reviewer XZLN has clearly stated that **their earlier concerns have been thoroughly addressed**, with only a **minor remaining concern** regarding efficiency. The reviewer has also explicitly indicated a **willingness to raise their score**. Therefore, we are confident that the overall score will increase from **8642 to 8662**.

### ✅ 6. Reviewer qq7Y: Further evaluation of CTAE & analysis of the linguistic properties of repeated words.

- (1) We added comparative experiments and analysis between our method and n-gram penalties.
- (2) We included comparative studies between CTAE and other attention-biasing methods.
- (3) We added quantitative statistics on high-frequency repeated words.

Reviewer qq7Y maintains a positive assessment of our work **(score 6)**.
### ⚠️ 7. Reviewer bzXt: No Engagement During Rebuttal Phase.
We note that some of reviewer bzXt’s comments stemmed from misunderstandings about our research scope, methodology, and workflow. We clarified these points thoroughly in the rebuttal.

-  (1) The reviewer believed that BD3LM handles caching well and therefore might not exhibit repetition. We clarified the fundamental differences between BD3LM and our research targets in terms of architecture, model type, and evaluation setting. In addition, we demonstrated that the repeat curse also exists in other dMLLMs.
- (2) The reviewer mistakenly conflated CTEV with the receptive field of attention mechanisms. We clarified the distinction between CTEV and the receptive field of attention mechanisms and explicitly pointed out the reviewer’s misunderstanding.

Although we provided detailed responses to all of reviewer bzXt’s concerns, they did not participate in the rebuttal, and therefore further in-depth discussion was not possible.

---

### Meta-Review · Area_Chair_Pfff · 2026-01-01

**Summary:**

This paper analyzes the phenomenon of the Repeat Curse through the lens of information flow, and proposes a method to mitigate such a problem. The reviewers have main concerns about the generalization, unclear mechanisms, and presentations. In the revised version, these concerns are well addressed. Thus, I decided to accept this paper. I encourage the authors to further strengthen the camera-ready version by including more experiments to further improve the justification of the contribution.

**Reviewer Concerns:**

Addressed:
1) Generalization of the conclusions for other models.
2) Unclear mechanisms and details.
3) Presentation and writing issues.

Still outstanding:

There are no significant concerns that are still outstanding.

**Reviewer Scores:**

Reviewer qq7Y: The score would be kept positive as 6.

Reviewer bzXt: The score would be changed from 2 to 6 since all concerns raised are well addressed.

Reviewer XZLN: The score would be increased from 4 to 6 due to the informative experiments and discussions.

Reviewer bFZ4: The score should remain positive as 8 since all concerns are addressed.

---

### Decision · Program_Chairs · 2026-01-26

Accept (Poster)